# A new bias-correction method for precipitation over complex terrain suitable for different climate states: a case study using WRF (version 3.8.1)

Patricio Velasquez[1,2], Martina Messmer[1,2,3], and Christoph C. Raible[1,2]

[1]Climate and Environmental Physics Institute, University of Bern, Switzerland
[2]Oeschger Centre for Climate Change Research, University of Bern, Switzerland
[3]School of Earth Sciences, The University of Melbourne, Melbourne, Victoria, Australia

*Correspondence to:* Patricio Velasquez (velasquez@climate.unibe.ch)

**Abstract.** This work presents a new bias-correction method for precipitation over complex terrain that explicitly considers orographic characteristics. This consideration offers a good alternative to the standard empirical quantile mapping method (EQM) during colder climate states in which the orography strongly deviates from the present-day state, e.g., during glacial conditions such as the Last Glacial Maximum (LGM). Such a method is needed in case that absolute precipitation fields are used, e.g., as input for glacier modelling or to assess potential human occupation and according migration routes in past climate states. The new bias-correction and its performance are presented for Switzerland using regional climate model simulations at 2-km resolution driven by global climate model outputs obtained under perpetual 1990 and LGM conditions. Comparing the present-day regional climate model simulation with observations, we find a strong seasonality and, especially during colder months, a height dependence of the bias in precipitation. Thus, we suggest a 3-step correction method consisting of (i) a separation into different orographic characteristics, (ii) correction of very low intensity precipitation, and (iii) the application of an EQM, which is applied to each month separately. We find that separating the orography into 400-m height-intervals provides the overall most reasonable correction of the biases in precipitation. The new method is able to fully correct the seasonal precipitation bias induced by the global climate model. At the same time, some regional biases remain, in particular positive biases over high elevated areas in winter and negative biases in deep valleys and Ticino in winter and summer. A rigorous temporal and spatial cross-validation with independent data exhibits robust results. The new bias-correction method certainly leaves some drawbacks under present-day conditions. However, the application to the LGM demonstrates to be a more appropriate correction compared to the standard EQM under highly different climate conditions as the latter imprints present-day orographic features into the LGM climate.

# 1 Introduction

The hydrological cycle is an important component in the Earth's climate system, because of its capability to transport and redistribute mass and energy around the world. Changes in the hydrological cycle can lead to droughts or floods and thus impact the ecosystem services. Moreover, it plays an important role in shaping the Earth's climate history (Mayewski et al., 2004). The latter is because the hydrological cycle shows a strong response to different external forcing functions and to changes in atmospheric compositions (Ganopolski and Calov, 2011; Stocker et al., 2013). Namely, hydrology and water resources are strongly influenced by changes in precipitation patterns (Stocker et al., 2013; Raible et al., 2016).

Cold periods, i.e., glacial periods, offer a unique testbed to better understand how the hydrological cycle responds to climate conditions highly different compared to today's climate. The Last Glacial Maximum (LGM) is the most recent glacial period and dates back to around 21 kya (Yokoyama et al., 2000; Clark et al., 2009). The LGM is characterized by large ice sheets in the Northern Hemisphere, a global mean temperature roughly 5 to 6.5 °C colder than today (Otto-Bliesner et al., 2006) and a global sea level of 115 to 130 m below the present-day one (Lambeck et al., 2014; Peltier and Fairbanks, 2006). Proxy records for Europe show that the climate was 10 to 14 °C colder and around 200 mm year$^{-1}$ drier during the LGM compared to recent climate conditions (Wu et al., 2007; Bartlein et al., 2011). These climatic conditions have strong implications not only for nature but also for humans. For instance, Burke et al. (2017) and Wren and Burke (2019) demonstrated the importance of climate conditions and its variability as drivers of human behaviour during the LGM, e.g., the spatial distribution of populations and influence on the cultural and biological evolution (Kaplan et al., 2016). Important modelling tools, e.g., global atmospheric climate models and hydrological models have been used to describe the Earth's system in the LGM. Compared to the sparse and local climate information from the proxies, these tools provide physically consistent and spatially gridded three dimensional information on various meteorological variables. Thus, they offer valuable information to improve the understanding of the responses and feedbacks to internal and external forcing on time scales longer than some centuries (e.g., Xu, 2000; Andréasson et al., 2004; Xu et al., 2005; Fowler et al., 2007a; Yang et al., 2010; Chen et al., 2012). Global climate models are generally in line with the proxy evidence and depict a European climate that was largely colder and drier than today. However, they underestimate the amplitudes of the changes compared to proxy evidence and poorly represent areas with complex terrain (e.g., Hofer et al., 2012a; Ludwig et al., 2016).

The modelling tools also show other uncertainties, in particular in the hydrological cycle, as not all relevant processes are explicitly simulated by the models (e.g., Ban et al., 2014; Giorgi et al., 2016). This is especially true for global models, which have a comparably coarse spatial resolution. Hence, most processes governing regional- to local-scale precipitation are not resolved and need to be parameterised (Leung et al., 2003; Su et al., 2012), resulting in a strong parameter dependence when simulating regional-scale precipitation (Rougier et al., 2009). To overcome some of the uncertainties, regional climate models (RCMs) are used to dynamically downscale global climate models. Many RCM simulations are carried out within the framework of the Coordinated Regional Downscaling Experiment (CORDEX), which defines one of the premier goals to better understand relevant phenomena at finer scales (Moss et al., 2010). Even though regional climate models can solve atmospheric equations on a much finer scale than global models, the simulated precipitation patterns still show large biases for

present-day climate when comparing them to observations. This has for example been illustrated by the CORDEX simulations analysed by Casanueva et al. (2016) and Rajczak and Schär (2017). These biases are not only produced by initial and boundary conditions provided by GCMs, but they are also related to regions characterized by complex topography and to processes that correspond to a finer scale, such as cloud microphysical processes. These processes need to be parameterised as they cannot be explicitly resolved because of the RCM resolution used in CORDEX (Boer, 1993; Zhang and McFarlane, 1995; Fu, 1996; Haslinger et al., 2013; Yang et al., 2013; Warrach-Sagi et al., 2013; Maraun and Widmann, 2015; Hui et al., 2016). To overcome these shortcomings, RCMs need to be run at a resolution where they can explicitly resolve some of the relevant processes, such as convection (e.g., Giorgi et al., 2016; Messmer et al., 2017). Even though the convection-resolving RCMs can describe precipitation much more precisely, biases are still evident (e.g., Ban et al., 2014; Gómez-Navarro et al., 2018). These inconsistencies and uncertainties may for example impact the results obtained through hydrological and glacier modelling that follow next in the modelling chain (Allen and Ingram, 2002; Seguinot et al., 2014; Felder et al., 2018).

Some climate change studies try to correct parts of these errors in precipitation patterns and intensities by so-called bias-correction methods (Maraun et al., 2010). These bias-correction methods are needed in case absolute values matter, e.g., for the forcing of impact models like glaciers or ice-sheets (Jouvet et al., 2017; Jouvet and Huss, 2019), when temperature thresholds are important as limiting factor for, e.g.,vegetation coverage, freezing of water, snowfall vs. rainfall, or when precipitation thresholds are essential (Liu et al., 2006; Zhao et al., 2017; Liu et al., 2018; Chen et al., 2019; Wang et al., 2020). So far, several correction methods are suggested in the literature, e.g., linear scaling, local intensity scaling, or power transformation (e.g., Berg et al., 2012; Fang et al., 2015; Lafon et al., 2013). An overview of different methods and their limitations are given in Maraun (2016) and Maraun and Widmann (2018b). Another important bias-correction method is the empirical quantile mapping (EQM) known as one of the best techniques to correct precipitation biases in the present-day climate (e.g., Lafon et al., 2013; Teutschbein and Seibert, 2012, 2013; Teng et al., 2015). If the method is applied to a climate state different from the present one, all these methods suffer basically from the assumption of stationarity in the biases, since they are trained with a climate that does not correspond to the simulated climate that is afterwards bias-corrected. Statistical relationships between observations and model output are used to estimate transfer functions in the observed period and are then applied to different climate states, e.g., past and future climate change scenarios. These statistical relationships and the bias structure can be altered by changes in the precipitation processes in the different climate states. Focusing on the LGM climate, an important process is related to changes in the albedo due to differences in vegetation and land cover (Kaplan et al., 2016; Velasquez et al., 2020). Also, changes in near-surface condensation processes may play an important role, i.e., freezing of near-surface moisture over areas covered by ice. These processes can influence the temperature and moisture profiles and thus also precipitation processes. Other important processes are linked to modifications in the general atmospheric circulation and in the water availability (Hofer et al., 2012b; Kageyama et al., 2020; Pinto and Ludwig, 2020). This can also regulate the water transport and thus, also the precipitation patterns.

Hence, these changes amongst others may violate the stationarity assumption of bias-correction methods. Besides the assumption of stationarity of the transfer functions, these correction methods only implicitly consider orographic features that strongly affect precipitation and its biases (e.g., Piani et al., 2010b; Amengual et al., 2011; Berg et al., 2012; Chen et al., 2013;

Cannon et al., 2015; Fang et al., 2015). Note that this implicit consideration relies on the orography where the method is trained on. Hence, the applicability of bias-corrections may not be justified to different climate states where the topography strongly changes such as in the LGM.

This calls for a flexible method that can ameliorate the assumption of stationarity in the biases when correcting precipitation
errors. One possibility is to apply a cluster analysis to precipitation and its biases to identify classes with similar bias behaviour. An example for Switzerland of such an approach is presented by Gómez-Navarro et al. (2018). The drawback of such an approach for our purpose is that the cluster analysis still relies on the characteristics and circulation of the current climate. To be as much independent from current climates as possible and to provide a correction that includes important characteristics of the Alpine climate, we came up with "static" characteristics, i.e. topography height and slope orientation and the assumption
that relationships to these static characteristics remain unchanged in different climate states. Thus, our work aims at presenting a new bias-correction method that fills this gap by using orographic features as variables for the correction. Such a correction avoids the explicit usage of current atmospheric circulation and provides a new alternative to the standard EQM for areas with complex topography during highly different climate states, i.e. glacial times.

The new method is based on EQM (Lafon et al., 2013; Teutschbein and Seibert, 2012, 2013; Teng et al., 2015) explicitly
combined with orographic characteristics, and attempts to correct wet or dry biases that are introduced by parameterisations and numerical formulations in global, regional or both models. Such biases include especially those that are associated with orographic effects, namely, vertical motion leading to precipitation. Observations or proxy reconstructions are limited over the Alps during glacial times. Thus, the method is directly evaluated under present-day climate conditions and its performance compared to the standard EQM is assessed in an LGM climate simulation. The data to be corrected stems from climate simula-
tion performed with the high-resolution RCM Weather Research and Forecasting (WRF) model (Skamarock and Klemp, 2008) driven by simulations under perpetual climate conditions using the Community Climate System Model version 4 (CCSM4, Gent et al., 2011). To estimate the transfer functions of the EQM we use two observation data sets, separately; one for Switzerland (MeteoSwiss, 2013) and one for the Alpine region (Isotta et al., 2014). The focus of the presented study is on the method itself and its evaluation. The latter consists of assessing the performance over the Alps, the temporal and spatial transferability,
and the comparison of the new method and standard EQM method (Lafon et al., 2013) under LGM conditions.

The paper is structured as follows. Section 2 describes the models and data sets used to construct the method. Section 3 presents the new bias-correction method. Section 4 evaluates the new method. Finally, a summary and conclusive remarks are given in Sect. 5.

## 2   Models and data

We use a present-day and an LGM simulation to create and evaluate the new bias-correction. Thereby, we employ a model chain that consists of a global climate model and a regional climate model, where the global climate model provides the boundary conditions for the regional climate model.

The global climate model is the Community Climate System Model (version 4; CCSM4; Gent et al., 2011). The model's atmospheric component is calculated by the Community Atmosphere Model version 4 (CAM4, Neale et al., 2010) and the land component by the Community Land Model version 4 (CLM4, Oleson et al., 2010). Only two components so-called data models are used for the ocean and sea ice, i.e., the atmospheric component is forced by time-varying sea surface temperatures and sea ice cover obtained from a coarser resolved fully coupled 1990 AD and LGM simulation with CCSM3, respectively (Hofer et al., 2012a). The atmosphere land-only model was run with a horizontal resolution of $1.25° \times 0.9°$ (longitude $\times$ latitude) and with 26 vertical hybrid sigma-pressure levels. Two global climate simulations are performed each covering 31 years: (i) under perpetual 1990 AD and (ii) under LGM conditions, respectively. The orbital forcing and atmospheric composition are adjusted to the respective period (Table 1). The temporal resolution of the output is 6-hourly. More detailed information on these simulations and their settings are presented in Hofer et al. (2012a, b) and Merz et al. (2013, 2014a, b, 2015).

To investigate the climate over central Europe and in particular over Switzerland in more detail, an RCM is used for the dynamical downscaling. Note that Switzerland is only covered by 12 grid points and the Alps are represented with a maximum height of approximately 1400 m a.s.l. in CCSM4. We use the WRF model version 3.8.1 for the dynamical downscaling (Skamarock and Klemp, 2008). The model is set up with four two-way nested domains with a nest ratio of 1:3. The domains have a horizontal resolution of 56, 18, 6 and 2 km, respectively, and 40 vertical eta levels. The outermost domain includes an extended westward and northward area that takes as midpoint the Alpine region (Fig. 1). Moreover, the innermost domain focusses on the Alpine region. The fine resolution of 2 km over this area is important as it covers a highly complex terrain. The resolution in the two innermost domains permits the explicit resolution of convective processes. Thus, no parameterisation for convection is used in these two domains and precipitation is described by microphysical processes (Table 2). Convection-permitting model resolutions are in general preferred as many recent studies show a better performance in simulating precipitation (e.g., Ban et al., 2014; Prein et al., 2015; Kendon et al., 2017; Berthou et al., 2018; Finney et al., 2019). However, we shall keep in mind that some biases in temperature and cloud formation may be produced by this set up, which may lead to additional biases in precipitation as shown in Ban et al. (2014). Table 2 lists the relevant parameterisation schemes chosen to run WRF with.

WRF is driven by, but not nudged to, the corresponding global simulation and is run for 30 years using perpetual 1990 AD and LGM conditions, respectively (Table 1). For the LGM simulation the surface conditions need some further adjustments. These include the lowering of the sea level and extended ice sheets as specified in the PMIP3 protocol (Fig. 3; for more details see: Ludwig et al., 2017). The glaciation over the Alpine region (obtained from Seguinot et al., 2018) and other glaciated areas (e.g., Pyrenees, from Ehlers et al., 2011) are modified according to LGM conditions (Fig. 3b). Additionally, the land cover and land use are altered to comply with LGM conditions, as described in Velasquez et al. (2020). Each 30-years simulation is split up into ten single 3-years simulations and carried out with adaptive time-step in order to increase the throughput on the available computer facilities. For each of the 3-years simulations, a 2-months spin-up time is considered to account for the longer equilibrium times of the land surface scheme of WRF. Tests show that the WRF land scheme reaches a quasi-equilibrium after approximately 15 days.

Two gridded observational data sets for daily precipitation are used: daily precipitation RhiresD (MeteoSwiss, 2013) and the Alpine Precipitation Grid Dataset (APGD; Isotta et al., 2014). Both data sets cover more than 35 years. In this study,

we use only the 30-years period 1979–2008. Note that we carry out a bilinear interpolation using the Climate Data Operators (CDO, Schulzweida, 2019) to convert both observational data sets into the corresponding grid of WRF. The RhiresD has a spatial resolution of approximately $2 \times 2$ km and covers only Switzerland (MeteoSwiss, 2013). This data set is based on rain gauge measurements distributed across Switzerland (for more details see; Isotta et al., 2014; Güttler et al., 2015). These

point measurements are spatially interpolated to obtain a gridded data set, which is described in more detail in Frei and Schär (1998), Shepard (1984) and Schwarb et al. (2001). The APGD encompasses the entire Alpine region with a spatial resolution of $5 \times 5$ km (Isotta et al., 2014). It was developed in the framework of EURO4M (European Reanalysis and Observations for Monitoring) by using a distance-angular weighting scheme that integrates climatological precipitation using the local orography and the rain gauge measurements (Isotta et al., 2014). For our analysis, the Alpine areas of Italy and Slovenia are

excluded from APGD because of their poor station density covering the period 1979 – 2008 compared to RhiresD, especially over complex topography and at high altitudes. Note that all data sets consider daily precipitation as total precipitation, i.e., both solid and liquid precipitation, and convective and non-convective precipitation. Moreover, days without precipitation are treated as censored values, i.e., not considered in the analysis, when daily precipitation is equal to 0 mm, although in the case of observations this is equivalent to 0.1 mm day$^{-1}$ due to gauge precision.

The observational gridded data sets provide valuable insights. However, they also contain some discrepancies and uncertainties due to inter- and extrapolation methods, e.g., high precipitation intensities are systematically underestimated and low intensities overestimated, especially in areas where observations are not available, i.e., on high elevated areas, such as mountain peaks. The magnitude of these errors depends on the season and the altitude. In regions above 1500 m a.s.l., the error can be higher than 30 % because of a "gauge undercatch" induced by strong winds and the applied interpolation method carried out

with a distance-angular weighting scheme (Frei and Schär, 1998; Nešpor and Sevruk, 1999; Auer et al., 2001; Ungersböck et al., 2001; Schmidli et al., 2002; Frei et al., 2003; MeteoSwiss, 2013; Isotta et al., 2014). Note that the limitations of the observational data sets are not included in the analysis of this study, i.e., we consider the observational gridded data sets as truth. Nevertheless, one shall keep the limitations of the observational data in mind, in particular when discussing the remaining biases in areas and seasons where the observational data sets also have problems.

## 25  3   Bias-correction

The correction method, developed in this study, consists of three steps: (i) separation with respect to different orographic characteristics, (ii) adjustment of daily precipitation with very low-intensity, and (iii) application of the EQM. Each of these three steps is described in more detail in the following paragraphs.

    In a first step, three orographic characteristics are used to separate the region of interest into several groups. These char-

acteristics are height, slope-orientations, and a combination of both. The height ranges from circa 200 m a.s.l. to a maximal value of 3800 m a.s.l. over the area of interest. Thus, the groups are selected by height-intervals, which cover the range from 400 to 3200 m a.s.l. Two height-intervals are tested separately: 100 or 400 m (e.g., height-intervals of 400 m are shown in Fig. 2). The heights below 400 and above 3200 m a.s.l. are considered as two additional height-intervals. The sec-

ond characteristic, used to group the region of interest, are four slope-orientations: north ( $315° \leq$ slope-orientation $< 45°$ ), east ($45° \leq$ slope-orientation $< 135°$), south ($135° \leq$ slope-orientation $< 225°$) and west ($225° \leq$ slope-orientations $< 315°$). Note that this characteristic is obtained by summing the two slope vectors that are directly provided by the RCM. Combining both characteristics, the groups are selected by height-intervals and then separated into sub-groups by the slope-orientations.

In a second step, we correct the daily simulated precipitation with very low-intensity in each group (or sub-group) and each month of the year, separately. The reason for this is that the frequency of precipitation with very low-intensity is often strongly overestimated due to the drizzle effect produced by the RCM (Murphy, 1999; Fowler et al., 2007b; Maraun et al., 2010).This overestimation can distort the precipitation distribution substantially, i.e., shifting the quantiles, producing inappropriate corrections in the third step when EQM is applied (Teutschbein and Seibert, 2012; Lafon et al., 2013).

To correct precipitation with very low-intensity, simulated precipitation values are censored by setting them to zero when they are below a specific threshold. Many studies use a static threshold for the entire simulated data set which is between 0.01 and 1 mm day$^{-1}$ (Piani et al., 2010a; Lafon et al., 2013; Maraun, 2013). To be consistent with the different biases-treatment across the groups, we calculate a static threshold for each group (or subgroup) and each month of the year. Thus, we carry out the first part of the local intensity scaling method (Schmidli et al., 2006; Teutschbein and Seibert, 2012) before applying

the quantile mapping technique. This method consists of choosing the threshold in a way such that the number of days with precipitation in the simulation coincides with the precipitation-day occurrence from the observations. In our work, the threshold can vary from group to group and from month to month between 0.001 and 1 mm day$^{-1}$.

    In a third step, we correct the daily precipitation rate using an EQM method (Themessl et al., 2011; Lafon et al., 2013; Fang et al., 2015; Teng et al., 2015). Note that censored values are excluded from this step. EQM is based on the assumption

that all probability distribution functions are unknown, i.e. non-parametric (Wilks, 2011). The method consists of adjusting the quantile values from a simulation ($Q_{sim}$) to those from the observations ($Q_{obs}$) through a transfer function (TF; Fig. 4). The method is implemented by splitting each cumulative distribution function, i.e., observed and modelled, into 100 discrete quantiles. For each quantile value, the adjustment is carried out with a linear correction (Lafon et al., 2013), where $Q_{sim}$ is transformed into $Q_{sim}{}^{*}$ (corrected quantile; Eq.1).

$$Q_{sim}{}^{*} = TF \times Q_{sim} \qquad , \text{where } TF = \frac{Q_{obs}}{Q_{sim}} \qquad (1)$$

This linear correction is akin to the factor of change or delta change used in Hay et al. (2000). For values that are between quantiles, the same linear correction is used, but the TF is approximated by using a linear interpolation between the TFs related to the two nearest quantiles. In cases where values are below (above) the first (last) quantile, the TF related to the first (last) quantile is used for the adjustment. Similar methods were successfully applied to correct biases in precipitation simulated by

RCMs (e.g., Sun et al., 2011; Themessl et al., 2012; Rajczak et al., 2016; Gómez-Navarro et al., 2018).

    To combine all steps, the first part of the local intensity scaling method and the EQM are applied to each (sub-) group defined in the first step and to each month of the year, separately, by pooling all grid points that belong to each group and handling them as a single distribution of daily precipitation. This results in a set of TFs for each (sub-) group and each month of the year. For instance, it results in nine TFs for each month and in total 108 TFs throughout the year when the correction is carried out

using height-classes of 400 m. Moreover, the correction is afterwards applied to the daily precipitation at every grid point using the TFs that are common to all elements within the same group (or sub-group) and month. Thus, the new correction method guarantees that seasonality and height are taken into account.

To come up with a final method for the Alpine region, we first evaluate the influence of the different orographic characteristics (step 1). To be consistent with former studies (e.g., Sun et al., 2011; Themessl et al., 2012; Wilcke et al., 2013; Rajczak et al., 2016), the evaluation uses the same region where the TFs are estimated. This means that the Swiss region in the WRF output (at 2-km resolution) is defined as the area to be corrected and RhiresD (at 2-km resolution) is used to obtain the TFs and to evaluate the different correction methods. These TFs are called Internal TFs (Int-TF) during the cross-validation process later on.

Once the final method is determined, we apply two cross-validations to test the method more rigorously as suggested by Bennett et al. (2014). First, a temporal cross-validation is applied. Thereby, the 30-years period is split into a 15-years training period and an independent 15-years verification period. New sets of TFs are calculated from the first and last 15 years of the 30-years period, separately. Each set of TFs is then applied to the first and last 15 years, which results in four newly corrected precipitation data sets; namely, two dependent and two independent ones. Second, we apply a spatial cross-validation. Thereby, Switzerland is defined as the area to be corrected (WRF output at 2-km resolution). For the spatial cross-validation, an additional set of TFs is then estimated from the corresponding Alpine region of Germany, France, and Austria excluding Switzerland (called External TFs; Ext-TF) using APGD (at 5-km resolution; Fig. 1c). Ext-TFs are carried out at 5-km horizontal resolution and applied to Switzerland at 2-km resolution. This guarantees that no additional uncertainty is introduced by a spatial interpolation when comparing the results of Ext-TF and Int-TF. To see that the coarser resolution of APGD has no influence on the result, the performance of the correction method is also evaluated when using Ext-TFs trained at 5-km and then applied to the Swiss region at 5-km resolution. Note that these results only show small differences to the 2-km results and are therefore not shown. To determine the improvement of the new method, we compare it to a simple method that is carried out without orographic features using one EQM for the entire region in each month (12 EQM in total, referred to one EQM-TF hereinafter).

## 4  Validation of the method

### 4.1  Biases of WRF and their seasonality

To obtain insights into the performance of the RCM over complex topography, we compare the spatial and temporal representation of the simulated precipitation (the raw model output) with RhiresD. Focusing on monthly mean precipitation intensity across Switzerland, the box plots illustrate biases in the climatological annual mean cycle (Fig. 5a). The climatological mean values are slightly overestimated during colder months, i.e., between November and March, and are underestimated during warmer months, i.e., between April and October, but especially in September. In addition to the climatological mean values, Fig. 5a also shows the distributions of monthly mean precipitation intensity and their interquartile ranges. In colder months, the simulated distributions are wider and shifted to higher values than the observed distributions, whereas a clear shift to less

precipitation is found compared to the observed ones during warmer months. Overall, the interquartile ranges are reasonably simulated, which means that WRF realistically represents the variability of monthly mean precipitation intensity. Extreme precipitation, however, is strongly underestimated.

The annual cycle and the distributions of monthly mean precipitation intensities are estimated for different height-classes to get additional understanding of the behaviour of the simulated precipitation and also to explicitly illustrate the relation of the precipitation biases to the topography. This is summarised in Fig. 5b and 5c for the height-classes 400–800 m and 2800–3200 m that mostly represent the low and high altitudes, respectively. The climatological monthly means of the colder months, i.e., from November to March, are generally underestimated in the lower height-classes but overestimated at high altitudes. Additionally, we assess the biases at each grid point in a scatter-plot. To that end, we select two months that mainly represent colder and warmer months; namely, January and July, respectively. We find a clear positive correlation between the biases and altitudes in January (6a). In warmer months, i.e., April to October, both height-classes 400–800 m and 2800–3200 m reveal an underestimation in the climatological monthly means compared to the observations. This is again confirmed by scatter-plots between biases at grid points and altitude, where only a mean shift is found in July (6b). Overall, the simulated annual cycle changes from a weak cycle at low altitudes, in agreement with the one of the observations, to a strong and inverse seasonal cycle at high altitudes (Fig. 5b and 5c). An inverse annual cycle is also identified by Gómez-Navarro et al. (2018), who used a similar model chain as in this study. These authors found that the inversed annual cycle in precipitation is caused by the driving global climate model. Furthermore, we observe positive biases in the interquartile ranges during colder months, and a slight underestimation during warmer months (Fig. 5b and 5c). So far, the analysis of the biases suggests that including the height dependence can help in improving correction methods.

To better describe the spatial biases related to colder and warmer months, we select two months that mainly represent each period; namely, January and July. For these example months, we present the spatial patterns of the biases in the monthly mean precipitation intensity, in the variability illustrated by the interquartile range, and in the wet-day frequency. Note that the observational data sets are generally considered reliable and represent orographic features well, although at high altitudes less observations are available (Isotta et al., 2014). Furthermore, these spatial patterns implicitly illustrate the relation between the precipitation biases and the topography considering an uncertainty of around 30 % acceptable in the simulated precipitation due to the uncertainty in the observational data sets (Sect. 2).

The biases in the climatological mean precipitation intensity at each grid point (Fig. 7a and 7d) confirms the height dependence and seasonality already shown in Fig. 5. The strongest positive biases are mainly observed over mountains and during colder months, whereas the Swiss Plateau seems to be reasonably well simulated (Fig. 7a). Note that also the observations tend to underestimate precipitation in mountain regions so that a part of the strong positive bias is related to observational uncertainties (Isotta et al., 2014). In warmer months, the strongest negative biases are found in the north-western part of Switzerland, Ticino and in the steep valleys, where the Rhone Valley is marked by the strongest biases. In high mountain regions, smaller positive biases are identified during warmer months than during colder months (Fig. 7d). The strongest biases over mountains and in steep valleys seem to be induced by an amplification of different observed precipitation climatologies that govern those areas; namely, the mountains are known as wet regions and the steep valleys as dry areas (for more details see; Frei and Schär,

1998; Schwarb et al., 2001). This gives a first hint that different processes may lead to the biases. The positive precipitation bias over mountains in colder months may be mainly related to wet bias of the global simulation and synoptic transport, which is also overestimated in the global simulation (Hofer et al., 2012a, b). The resolution of the RCM seems to be important as this affects the representation of steep valleys, especially during convective processes in warmer months. The same is also true for

colder months, but to a lesser extent, as convective processes only play a minor role in these months.

    The interquartile ranges of the distribution of monthly mean precipitation intensity at each grid point (Fig. 8a and 8d) are strongly overestimated over the Alps during colder months, whereas they are generally smaller compared to the observations during warmer months. The biases are stronger than the ones observed in the climatological mean value (Fig. 7a and 7d), which means that the variability simulated by WRF is strongly season-dependent (Fig. 8a and 8d). The increase in variability during

colder months is a hint that processes common during winter, e.g., the synoptic atmospheric systems, may be too efficient in producing precipitation compared to the observations. The reduced variability in warmer months hints to remaining problems in convective processes as these are more relevant during summer. Also, observations do not perfectly estimate the range due to their uncertainty that fluctuates from 5 % over the flatland regions to more than 30 % in high altitudes (Isotta et al., 2014).

    Another important measure to characterise precipitation is the occurrence of precipitation at each grid point, defined by the

wet-day frequency (the number of days with precipitation rate of at least 1 mm day$^{-1}$). The wet-day frequency is strongly overestimated during colder months, but shows only a slight overestimation during warmer months (Fig. 9a and 9d). This overestimation can be also related to the well-known problem in regional climate modelling, i.e., the simulation of a higher frequency in precipitation but at the same time with a lower intensity than observed (Murphy, 1999; Fowler et al., 2007b; Maraun, 2013). The overestimation in wet-day frequency, so-called drizzle effect, can be mainly related to the occurrence

of synoptic atmospheric systems commonly observed during colder months and not to local convective processes that are frequently observed during summer (for climatology see Frei and Schär, 1998; Isotta et al., 2014). Furthermore, the positive bias in the wet-day frequency may slightly contribute to the underestimation of the extreme precipitation (Fig. 5) as precipitable water, which is necessary for extreme precipitation events, is removed via the drizzle effect. Namely, the precipitable water available for a daily extreme precipitation event is distributed over several days due to problems in the parameterisations of the

cloud microphysical and precipitation processes as found in Knist et al. (2018).

## 4.2   Influence of different orographic characteristics on the performance of the bias-correction method

    Different orographic characteristics are suggested to be used as classification in the new bias-correction method (step 1 in Sect. 3): the height-intervals (100 m and 400 m), the slope-orientations, and a combination of both using the height interval of 400 m (combined-features). Note that the results are not affected by interchanges in the order of the orographic characteristics

in the combined-features (therefore not shown). We assess in the following, which of these characteristics are necessary to improve a simple approach of applying one EQM-TF to the entire domain, where orographic features are not considered. An improvement compared to one EQM-TF for the entire domain would certainly support the height dependence of the biases. Note that we do not compare our results to the standard EQM as the latter would outperform the here described method by definition. Note that the standard EQM removes the mean bias on a grid-point level as it is a statistical downscaling at the

same time. We use Taylor diagrams (Fig. 10) for four months namely January, April, July, and September, as the biases show a strong seasonality (see previous section). The evaluation is carried out with three statistics: the spatial correlation, the spatial root-mean-square-error and the spatial standard deviation.

Figure 10a shows that the correction methods using height-intervals of both, 100 and 400 m, and the combined-features have a better performance during the colder months than the other methods, i.e., using just orientation or one TF for the entire domain: the standard deviation is better adjusted, especially when using height-intervals of 100 m, the root-mean-square-error is reduced by roughly 32 %, and the correlation is slightly increased (Fig. 10b). During the cold-to-warm transition months (here illustrated by April), the correction using height-intervals of 400 m and the combined-features have a better performance than the other settings. This is because the standard deviation is fully adjusted, the root-mean-square-error is reduced by 17 %, and the correlation is increased to $r = 0.75$ (Fig. 10b). During the warmer months, all correction methods except the one using height-intervals of 100 m show a similar good performance, i.e., the standard deviation is fully adjusted, the root-mean-square-error is slightly reduced, and the correlation is slightly increased (Fig. 10c). The similar good correction in the warm month can be explained by a reduced height dependence of the biases in these months. During the warm-to-cold transition months (September, Fig. 10d) all correction methods show a similar performance increase compared to the observations, correlation and root-mean-square-error are only slightly improved. The method using height-intervals of 100 m often reduces the standard deviation. This can be explained by a reduced data coverage which means less variability within some height classes as a smaller climatological range is encompassed by each height-interval.

Even though, all the settings mostly show a good performance, the one using height-intervals of 400 m outperforms in most measures and months. In addition, the correction method using the height-intervals of 400 m needs less computational time compared to the similarly good correction method using height-intervals of 400 m and slope-orientations. Therefore, the method using height-intervals of 400 m seems to be the most appropriate setting and is used in the following analysis.

## 4.3 Application of the bias-correction method and cross-validation under present-day conditions

The bias-correction method using height-intervals of 400 m is now assessed in more details. First, we focus on results where the TFs are estimated in the domain of Switzerland using 30 years (Int-TFs). Second, we discuss the results obtained by a temporal and spatial cross-validation technique, i.e., the TFs trained on another period and the TFs estimated with the surrounding Alpine region, excluding Switzerland (Ext-TFs). As in Sect. 4.2, a comparison to the standard EQM (Lafon et al., 2013) is not presented, since the standard EQM outperforms the new method under present-day conditions. A priori, this comparison is based on different prerequisites, as the standard EQM corrects at a grid-point level and thus, it removes the mean biases as in statistical downscaling methods. Instead, we again compare the new method to a simple one EQM-TF used for entire Switzerland. A similar approach is sometimes used in other studies as well to assess the added value of their proposed methods (e.g, Gómez-Navarro et al., 2018; Casanueva et al., 2016).

To illustrate the improvement by the correction method using Int-TFs, we compare the spatial and temporal representation of the corrected precipitation with RhiresD. Focusing on the monthly mean precipitation intensity across Switzerland, we find that the climatological annual cycle of mean precipitation intensity fully coincides with the one of the observations (Fig. 5a).

Also, the distributions of monthly mean precipitation intensity are fully adjusted and the corresponding interquartile ranges mainly correspond to the ones of the observations when using the new bias-correction method. Still, the extreme precipitation events are underestimated with the new method, which is expected as the TF of the extreme values is poorly constrained in the EQM approach (e.g., Themessl et al., 2011). The segregation into the height-classes (Fig. 5b and 5c) shows that the climatological monthly means and the distributions of monthly mean precipitation intensity are also well adjusted compared to the observations. This illustrates that the bias-correction method using height-intervals of 400 m is appropriate.

To further describe the spatial improvements of the new bias-correction method, we select here, as in the Sect. 4.1, two months that mainly represent the colder and warmer months, e.g., January and July, respectively. We again focus on biases in the monthly mean precipitation intensity, in the variability illustrated by the interquartile range, and in the wet-day frequency.

A comparison between Fig. 7a and 7d with Fig. 7b and 7e, shows that the biases in the climatological mean precipitation intensity are substantially reduced, especially the overestimation over high mountain regions during colder months and the general underestimation during warmer months. Still, regions with positive and negative biases remain over the eastern part of the mountains in colder months and in the steep valleys like the Rhone Valley in warmer months. Also, the negative bias in the Ticino during colder months remains, albeit it is slightly ameliorated. The rather moderate performance in these regions can be traced back to the fact that some height-classes sample over regions with different biases. Hence, biases of one area are diminished by the biases that are shared by the other areas. For instance, the strong negative biases observed in the Rhone Valley and Ticino are not fully decreased because the slight underestimation from the Swiss Plateau dominates this height-class (Fig. 7b and 7e).

To assess the improvements with respect to precipitation variability, we focus on the interquartile range of the distribution of monthly mean precipitation intensity at each grid point (Fig. 8b and 8e compared to Fig. 8a and 8d). The biases of the interquartile range improve only moderately, i.e., the strong overestimation over the mountains is partly corrected during colder months but not during warmer months. The underestimation over the flatlands and steep valleys is corrected during warmer months and poorly during colder months.

For the wet-day frequency, we find that the positive biases are mostly reduced, especially the strong overestimation over the mountains during colder months (Fig. 9b and 9e). However, the regions of Rhone Valley and Ticino, which show no biases in the raw model output, are slightly underestimated during colder months. The negative biases observed in the region of Grisons become stronger during colder months and in the region of Rhone Valley during warmer months (Fig. 9b and 9e). This effect is again caused by sampling different regions with different biases in the height classes.

Recent studies by Maraun et al. (2017) and Maraun and Widmann (2018b) showed that the observational and simulated data sets do not have a synchronised internal climate variability and, thus, this may be one of the sources of the remaining biases in free-running models. To assess these remaining biases, we perform a temporal cross-validation. An option could be to carry out a one-leave-out verification method to hold back most of the years for calibration; however, different lengths between calibration and the independent verification periods can lead to more uncertainties (Lafon et al., 2013; Maraun, 2016; Maraun et al., 2017; Maraun and Widmann, 2018b). Therefore, our temporal cross-validation consists of using different same-length periods for the calibration and the verification (see Sect. 3). Overall, the bias-correction method performs similar in

the independent 15 years and shows similar remaining biases as when using the entire 30 years for training and verification. Still, some differences between dependent and independent periods are evident: During January, the method trained on the first 15 years and verified in the second 15 years shows less biases over high altitudes and slightly higher biases in the flatlands and in the Ticino (not shown). Inversely, the method trained with the second 15 years and verified in the first 15 years shows

reduced biases in the flatlands and in the Ticino but not over the mountains (not shown). During July, similar small differences are identified in the independent verification periods (therefore not shown). Thus, there is a potential that a different internal climate variability affects the bias-correction method (Maraun et al., 2017; Maraun and Widmann, 2018b). However, these differences can be considered minimal as the accuracy of bias-correction methods is sensitive to the length of the period the methods are trained on (a shorter training period results in a less accurate performance; Lafon et al., 2013).

To further check the robustness of the new bias-correction method, a spatial cross-validation is performed (see Sect. 3). Thereby, we apply the TFs estimated from an independent data set of the Alpine region (at 5-km resolution) excluding Switzerland (Ext-TFs) to the Swiss region (at 2-km resolution). To have insights into the effects of the correction method using Ext-TFs, we compare the spatial and temporal representation of the corrected precipitation with the results obtained by the Int-TFs. Note that the RhiresD is always used as observations for the bias calculation. Again, to describe the spatial effects, we

select here two months that mainly represent the colder and warmer months, i.e., January and July, respectively.

A comparison between Fig. 7b with 7c shows almost the same pattern, i.e., the improvement in mean precipitation achieved by using Ext-TFs is similar to the Int-TFs during colder months. Still, some positive biases over the mountains seem to be smaller using Ext-TFs than Int-TFs, whereas the remaining negative biases are slightly stronger than the ones after using Int-TFs (Fig. 7b and 7c). The reason for the latter could lie in the inclusion of larger regions in the north and west of the

Alps mixing different climate conditions and thus bias behaviours. The slightly better performance in the mountain regions is probably related to more data available in these height classes, i.e., more grid-points at high altitudes (Fig. 2), and thus it is possible to better constrain the TFs. In the warmer months, we find that the method using Ext-TFs shows slightly more negative biases than with Int-TFs, in particular over the Swiss plateau. Again, we hypothesise that the inclusion of larger regions in the north and west of the Alps is responsible for this bias behaviour.

The interquartile ranges of the distribution of monthly mean precipitation intensity are similar when using either Ext-TFs or Int-TFs for the colder months (Fig. 8c compared to 8b). During warmer months, the negative biases in the western part of Switzerland are less improved using Ext-TFs than Int-TFs, again a hint that the inclusion of larger regions in the north and west of the Alps in the lower height-classes plays a role in the bias of the interquartile range.

The wet-day frequencies are very similarly corrected as in the approach using Ext-TFs compared to Int-TFs (Fig. 9c and 9f

compared to Fig. 9b and 9e). Thus, the wet-day frequency seems to be insensitive to the region where the TFs are estimated from.

Additionally, to further assess the local improvements of adding topographic features into the correction, we analyse the remaining biases of the simple method using TFs deduced for the Swiss region (Int-TFs), as described in Sect. 4.2, and for the corresponding Alpine region (Ext-TFs), separately. Overall, the comparison between the simple method and the new method

shows small differences (therefore not shown). The new method shows a better performance than the simple method in January

but a similar performance in July. Furthermore, the simple method increases the original biases over the flatlands, which are reduced by the new bias-correction. This confirms the results of the Taylor diagram illustrated in the Fig. 10, i.e., the better performance of the method using height-intervals of 400 m.

In summary, the new correction method reasonably well corrects biases in the monthly mean precipitation intensity, in the variability illustrated by the interquartile range, and in the wet-day frequency. The two cross-validations show that the improvements achieved by the new method are almost independent of the time period and region used to estimate the TFs. Additionally, the new method outperforms the simple method (one EQM-TF) in the present-day climate.

## 4.4 Application of bias-correction methods on the simulated LGM climate

To further examine the performance and applicability of the new bias-correction method, we apply it to the simulated LGM climate. Similarly, the standard EQM (e.g.; Lafon et al., 2013; Teutschbein and Seibert, 2012, 2013; Teng et al., 2015) is applied and precipitation fields resulting from its correction are compared to the one of the new method. The reason is that the strength of the standard EQM (correction at grid-point level) under present-day climate might be a weakness under highly different climate states, since local-related biases might not exist. To that end, we again focus on the monthly mean precipitation intensity over Switzerland in January and July, i.e., the two months that represent the cold and warm seasons, respectively.

Focusing on the raw LGM simulation first, we find wetter conditions in the southern part of the Swiss Alps (Fig. 11a and 11d) rather than at the north-facing slopes as it is the case in present-day conditions (more details about present-day conditions: Frei and Schär, 1998; Schwarb et al., 2001). Becker et al. (2016) indicated a strong precipitation gradient between the north- and the south-facing slopes in order to obtain a reasonable extent of the Alpine glacier during the LGM. This suggests an increase of intensity or frequency of the southerly moisture advection over the Alps. Also, Florineth and Schlüchter (2000) and Luetscher et al. (2015) indicated a circulation change from dominant westerlies in the present day to a more southerly atmospheric circulation during the LGM. From this brief qualitative analysis, we can conclude that WRF reasonably simulates the precipitation patterns during the LGM, even if the total amount might present some uncertainties.

Before assessing the performance of the two bias-correction methods, it is worthwhile to shortly focus on the changes in the topography. The comparison between the present-day (Fig 2) and LGM (Fig 3b) topography shows that the topography is differently lifted across Switzerland during the LGM. While the mountainous areas become larger, the height of their peaks hardly changes. The present-day valleys are filled by ice during the LGM and thus, the deep valleys almost disappear. For instance, the Rhone valley exhibits a continuous slope towards its spring (Fig. 3b), while it is a narrow and deep valley with almost a constant elevation in the present-day topography (Fig. 2). Since the Alps were covered by ice, the fine and complex present-day topography is lacking during the LGM.

We apply the standard EQM and the new method to not only assess their performance but also to identify the strength and weakness of each method. Comparing Fig. 11b and 11e to Fig. 11c and 11f illustrates that the corrections do not modify the north-south precipitation gradient observed in the raw simulation (Fig. 11a and 11d). The standard EQM method (Fig. 12b and 12d) shows that the shape of the valleys and the mountain peaks of the present-day topography are imprinted on the raw LGM climate (Fig. 12a and 12c). The standard EQM seems to add a fine and complex structure to the precipitation pattern.

This complexity is hardly justified over the Alps during the LGM, as stated before, which suggests that adding this structure is unnecessary. The imprint of the present-day topography is related to the nature of the standard EQM that trains the TFs point-wise assuming static orographic features. The new correction method follows by definition the LGM topography showing a smoother correction for the LGM climate, which provides precipitation patterns that more appropriately represent the LGM situation. Proxy records could give an idea on the LGM precipitation amounts but there is a very limited number of them in Switzerland; thus, a more rigorous analysis of the application of the two methods to the LGM climate is not possible. However, the difference between the two methods demonstrates that the application of the new bias-correction is better suited than the standard EQM. Therefore, we consider it as more appropriate for climate states with strongly altered topography compared to today.

## 5 Summary and conclusions

In this study, we present a new bias-correction method for precipitation over complex topography, which takes orographic characteristics into account. This method is mainly designed for climate states where the topography is distinctively different to the present-day one, i.e., glacial times. This is particularly important for studies where absolute values of precipitation are essential, such as glacier and ice sheet modelling (Seguinot et al., 2014; Jouvet et al., 2017; Jouvet and Huss, 2019) and the assessment of human behaviour during glacial times (Burke et al., 2017; Wren and Burke, 2019). To illustrate the performance of the new method, two regional climate model simulations are performed with WRF at 2-km resolution over the Alpine region. We particularly focus on the performance over Switzerland.

The comparison between the WRF simulation and the observations over Switzerland shows that the biases are season dependent and related to the complexity of the topography, especially in colder months (November to March). These months exhibit positive biases over mountains and negative biases in steep valleys, whereas negative biases dominate during the warmer months (April to October), especially in the Rhone Valley and Ticino. Parts of the biases are introduced by the driving global climate model, in particular the seasonal biases (Gómez-Navarro et al., 2018). Moreover, the large scale atmospheric circulation of the global climate model is too zonal – a known problem in many models (e.g., Raible et al., 2005, 2014; Hofer et al., 2012a, b; Mitchell et al., 2017) – which cannot be fully compensated for by the RCM. Thus, the wet bias present in the global simulation (Hofer et al., 2012a, b) may be transported into the regional model domain rendering especially the colder months with more precipitation. Still, observations are also not perfect and underestimate precipitation in particular in high altitudes by up to 30 % (Isotta et al., 2014). Other biases are potentially induced by the RCM, e.g., a WRF simulation using a similar setting but driven by ERA-Interim (Gómez-Navarro et al., 2018) shows also a comparable overestimation of precipitation over mountain regions as the simulation used in this study. In addition, we find that the extreme precipitation values are underestimated. This is due to the drizzle effect (Murphy, 1999; Fowler et al., 2007b) that can remove moisture needed for the extreme precipitation, which mainly comes from physical parameterisations of the model itself (Solman et al., 2008; Menéndez et al., 2010; Gianotti et al., 2011; Carril et al., 2012; Jerez et al., 2013). A hint for this is given by the fact that the wet-day frequency in the simulation is enhanced compared to the observations.

Numerous approaches to correct biases exist (e.g, Maraun, 2013; Teng et al., 2015; Casanueva et al., 2016; Ivanov et al., 2018); nevertheless, they assume stationary orographic features that are then imprinted onto the other climate state when applying the correction. Hence, an alternative method is needed, which reduces this assumption so that it adds value to especially colder climate states characterised by a strongly changed topography, such as the LGM. The new method consists of three steps: the orographic characteristics differentiation, the adjustment of very low precipitation intensities, and the EQM. Different orographic characteristics, i.e., the height-intervals, the slope-orientations, and the combination of both, are tested showing that the method using height-intervals of 400 m is generally the most skilful correction compared to other orographic characteristics and at the same time it is computationally the most efficient one. In the colder months, the new method outperforms the simple method of applying one EQM-TF that is deduced for the entire region of interest and does not consider any orographic features.

Applying the new bias-correction method to the Swiss region exclusively shows that the biases are mostly corrected. In particular, the distribution of the monthly precipitation across Switzerland is mainly adjusted, the mean precipitation biases are substantially reduced, and the biases in the wet-day frequency are mostly reduced. The method better corrects the positive biases during colder than warmer months, and reversely, the negative biases during warmer than colder months. However, some biases are still observed, which is explained by the fact that some height-classes sample over regions with different biases. Also, the deficient constraint of the TFs in uttermost quantiles poorly corrects extreme values, i.e., below the first quantile and above the last quantile. Furthermore, part of the remaining biases may also be interpreted as possible error propagation, which initially comes from the interpolation methods and "gauge undercatch" in the gridded observational data sets, especially at high altitudes where less data is available (for more details see; Sevruk, 1985; Richter, 1995; Isotta et al., 2014).

The new method is temporally and spatially cross-validated. The 30-years period is split in a 15-years training and a 15-years independent temporal verification part. The results are similar to the case when the TFs are trained on and applied to the 30-years period. Still, such a cross-validation might be problematic as the method's accomplishment relies on the biases caught during the period the method is trained on, i.e., the asynchronism in the internal climate variability of the data sets (Maraun et al., 2017; Maraun and Widmann, 2018a). Maraun and Widmann (2018a) argued that cross-validation methods shall compare the correction with the observations on different climate states, i.e., the future or past climate state, otherwise they can produce false positive or true negative results. To overcome some of these possible limitations, we apply a spatial cross-validation that checks the transferability of the bias-correction method to a different climate state. We use an independent data set of the Alpine region (APGD) excluding Switzerland when estimating the transfer functions (Ext-TFs). This shows a similar improvement as the correction performed with data over the Swiss region exclusively (Int-TFs).

The applicability of the new method is further assessed under LGM climate conditions. There is a very limited amount of proxy evidence in Switzerland for a rigorous evaluation. Thus, we compare the performance of the new bias-correction method and the standard EQM when they are applied to LGM climate conditions. The standard EQM adds features to the precipitation that can be hardly justified in the LGM, whereas the performance of the new method suits better. This indicates that the new method is safer and therefore more appropriate than the standard EQM under LGM climate conditions. In a similar manner, the new method may also be better suited in some regions for future climate scenarios. This is especially true for areas that

are currently covered by ice, such as the Himalayas, since possible melting of glaciers can change the shape of the already complex terrain in the future.

Finally, a common drawback of all bias-correction methods (including the one presented in this study) is that they ignore a potential modification of the bias structure due to the handling of rainfall and snowfall in the model's microphysics. This is certainly important when the bias-correction method shall be used in cold climate states, like the LGM. Currently, there are no gridded and homogenised observations available for snowfall, which is needed for a rigorous analysis of this effect. Still, our seasonally separated and height-dependent method implicitly includes some aspects of the handling of rainfall and snowfall, since one can expect that most of the precipitation is snow at high altitudes and in colder months. Clearly, future work is needed on this aspect as soon as reliable observations of snowfall are available. Additionally, other variables of the Earth's system need to be assessed in future studies on bias-correction methods, especially the response of soil-moisture and snow-albedo to the corrected precipitation patterns. In the meantime, glaciologists can benefit from a better accuracy of precipitation data obtained by the new method for, e.g., LGM conditions. Glacier modelling (Seguinot et al., 2014; Jouvet et al., 2017; Jouvet and Huss, 2019) results may provide an alternative method for the validation when evaluating the prediction and proxy data of the glacier extents.

*Code and data availability.* WRF is a community model that can be downloaded from its web page (http://www2.mmm.ucar.edu/wrf/users/code_admin.php). The two climate simulations (global: CCSM4 and regional: WRF) occupy several terabytes and thus are not freely available. Nevertheless, they can be accessed upon request to the contributing authors. The post-processed daily precipitation that is used to perform the bias-correction is archived on Zenodo (Velasquez et al., 2019). The RhiresD and APGD can be requested from MeteoSwiss. Simple calculations carried out at a grid point level are performed with Climate Data Operator (CDO, Schulzweida, 2019) and NCAR Command Language (NCL, UCAR/NCAR/CISL/TDD, 2019). The figures are performed with NCL (UCAR/NCAR/CISL/TDD, 2019) and RStudio (RStudio Team, 2015). The codes to perform the bias-correction, the simple calculations and the figures are archived on Zenodo (Velasquez et al., 2019).

*Author contributions.* PV, MM, and CCR contributed to the design of the experiments. PV carried out the simulations and wrote the first draft. All authors contributed to the internal review of the text previous to the submission.

*Competing interests.* The authors declare no competing interests.

*Acknowledgements.* This work is supported by the Swiss National Science Foundation (grant 200021_162444). MM acknowledges support by the SNF (Early Postdoc.Mobility). The CCSM4 and WRF simulations were performed on the supercomputing architecture of the Swiss National Supercomputing Centre (CSCS). Thanks are due to European Reanalysis and Observations for Monitoring for providing APGD.

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

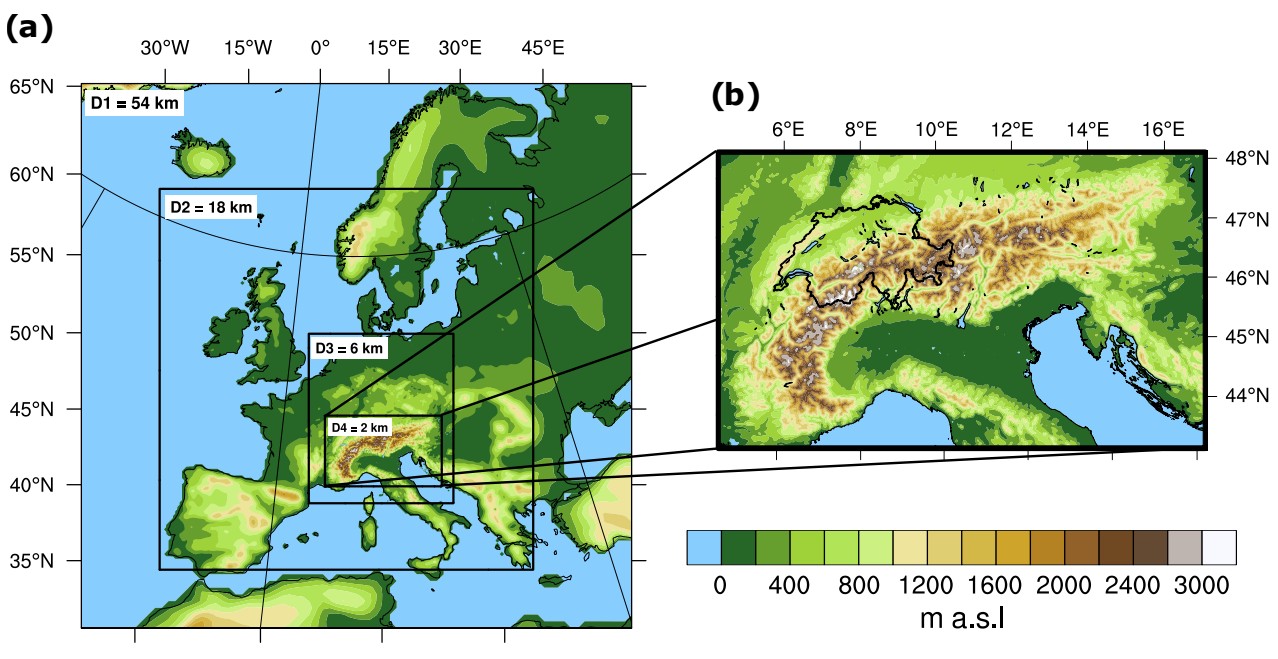

**Figure 1.** WRF domains and present-day topography. (a) illustrates the present-day topography and the four domains used by WRF. (b) shows the fourth domain including the area of interest (Switzerland) outlined by a black line.

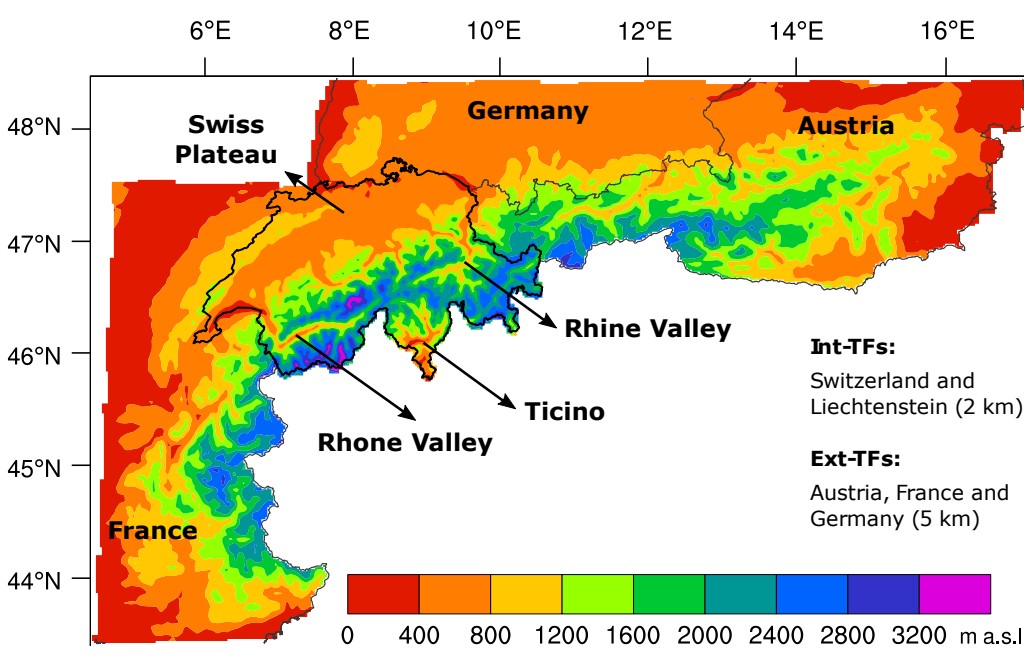

**Figure 2.** WRF innermost domain indicates the present-day height-classes used for the correction method (400 m interval) for the Int-TFs at 2-km resolution (Switzerland, black outline) and for the Ext-TFs at 5-km resolution (other shaded areas). Additionally, some labels are added to identify some specific areas in Switzerland that are used throughout the paper.

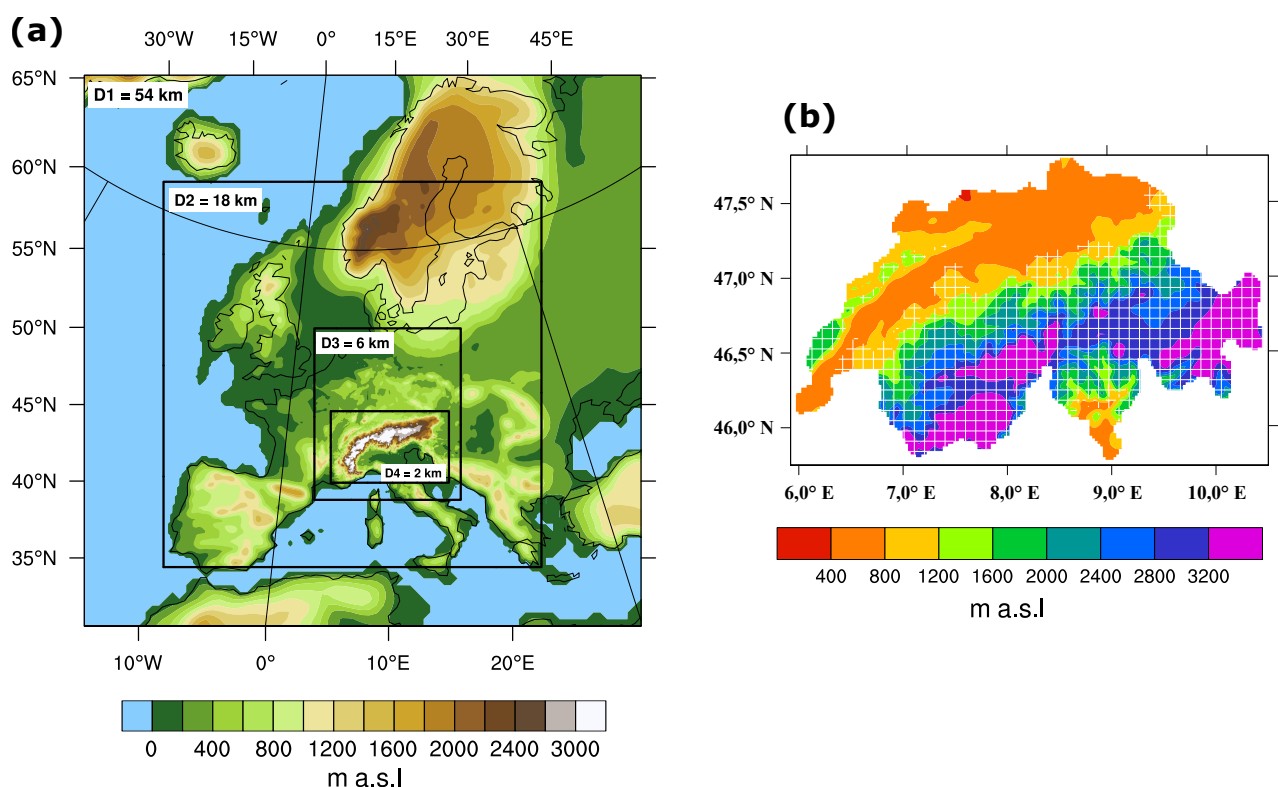

**Figure 3.** WRF domains and LGM topography. (a) illustrates the LGM topography, LGM sea level and the four domains used by WRF. (b) indicates the height-classes for the correction method (400 m interval) using the LGM topography over Switzerland at 2-km resolution, crosshatched areas are covered by glaciers.

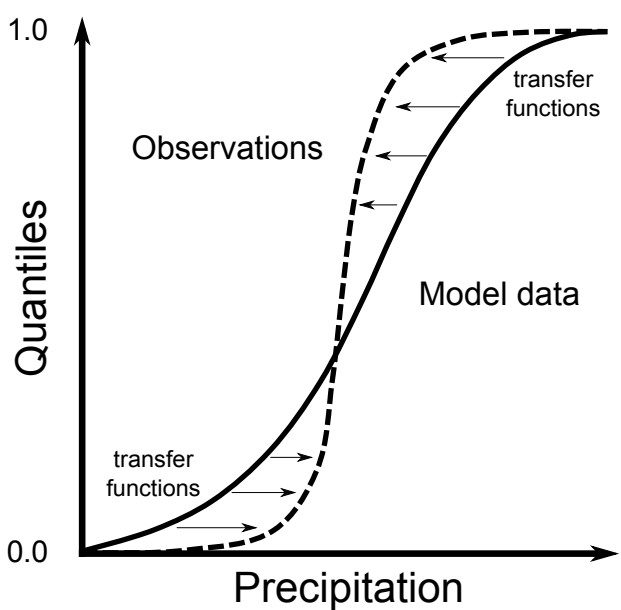

**Figure 4.** Diagram of the Empirical Quantile Mapping technique (EQM). Solid (dashed) line shows a schematic simulated (observed) cumulative distribution.

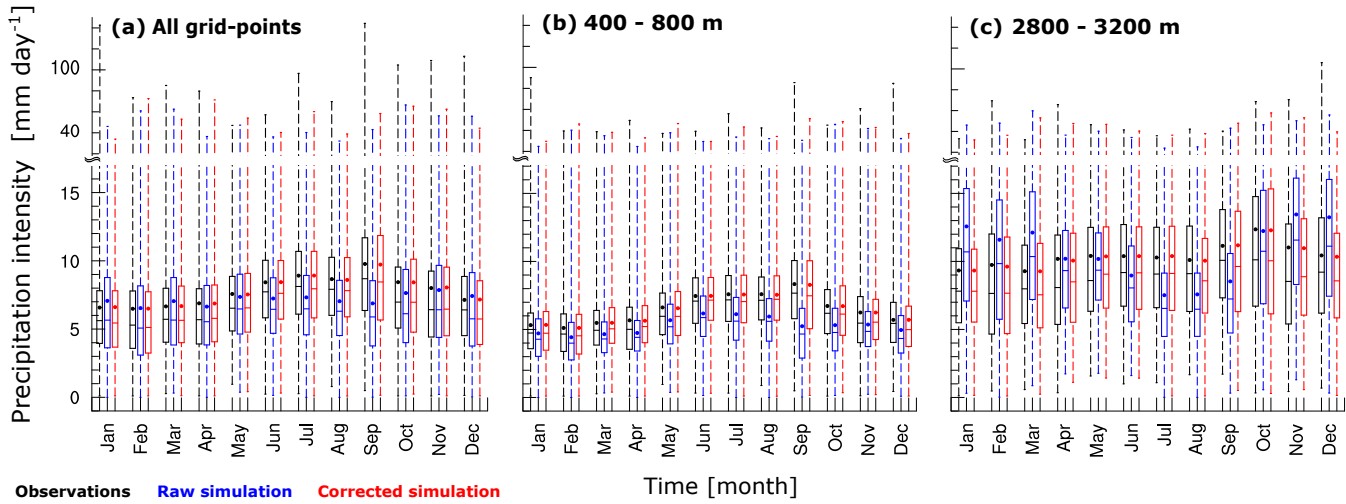

**Figure 5.** Boxplots illustrate the spatial distribution of monthly mean values of precipitation intensity across a specific area within 30 years: (a) the area covers all grid points over entire Switzerland, (b) the grid points in the height-class of 400 – 800 m, and (c) the grid points in the height class of 2800 – 3200 m. Black box-plots represent the observations (RhiresD), blue and red ones the raw and corrected simulation, respectively. Top and bottom ends of the dashed lines represent the maximum and minimum values, respectively. Dots represent the spatial climatological mean value.

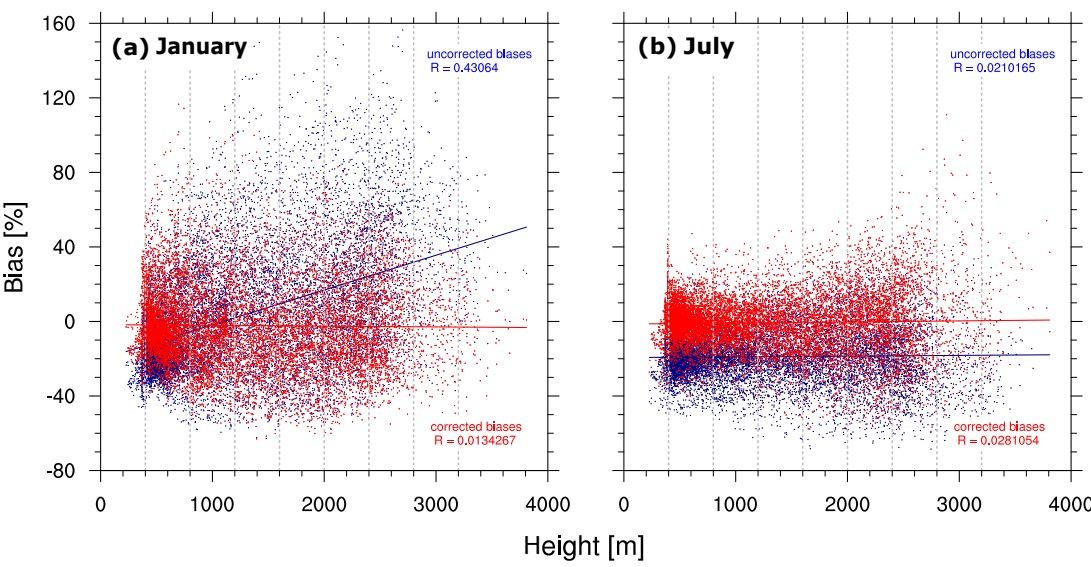

**Figure 6.** Biases over Switzerland. Blue (red) indicates the original (corrected) biases. (a) illustrates the bias versus height at each grid-point during January, (b) as (a) but for July. Solid lines represent the linear regressions, R the correlation between biases and height, and vertical dashed grey lines the boundaries of the height classes.

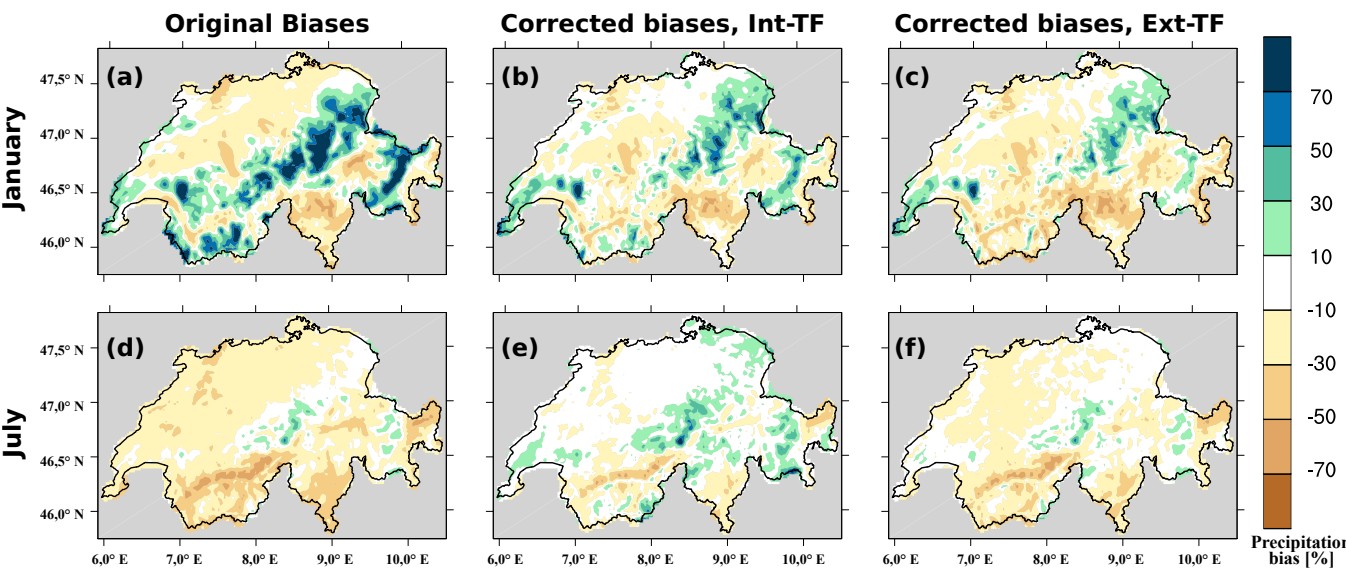

**Figure 7.** Biases in the climatological mean value of precipitation intensity over Switzerland. (a) represents the original biases in January, (b) the biases after being corrected using Int-TFs in January, (c) the biases after being corrected using Ext-TFs in January, (d), (e), and (f) as (a), (b), and (c) but in July, respectively.

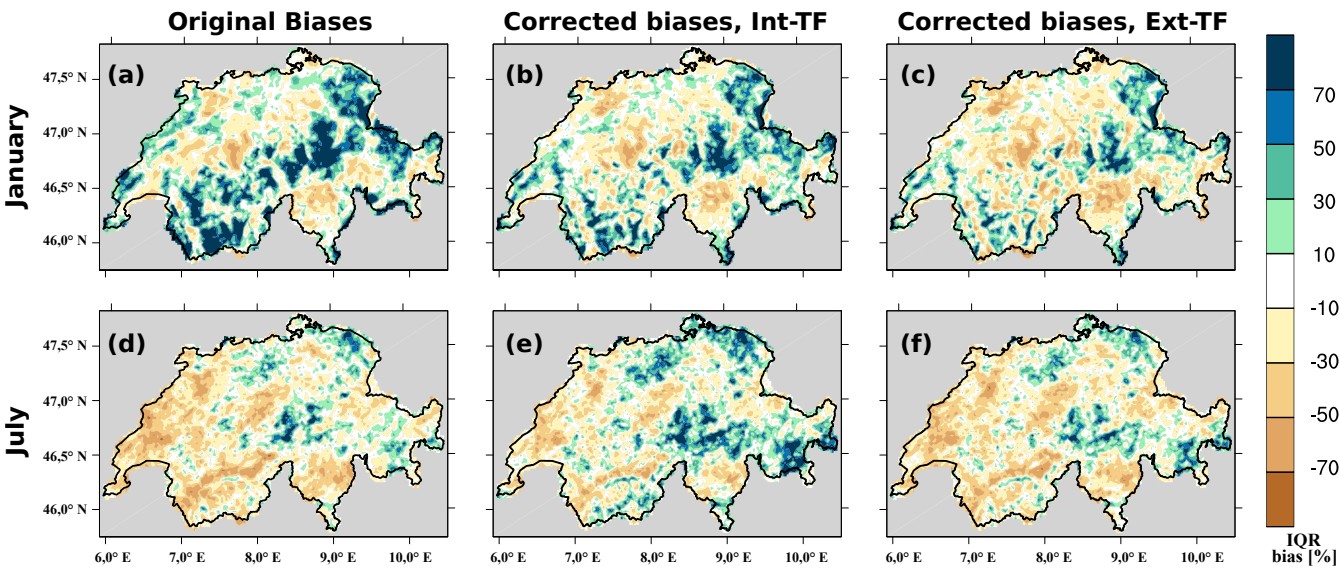

**Figure 8.** Biases in the interquartile range of monthly mean precipitation intensity over Switzerland. (a) represents the original biases in January, (b) the biases after being corrected using Int-TFs in January, (c) the biases after being corrected using Ext-TFs in January, (d), (e), and (f) as (a), (b), and (c) but in July, respectively.

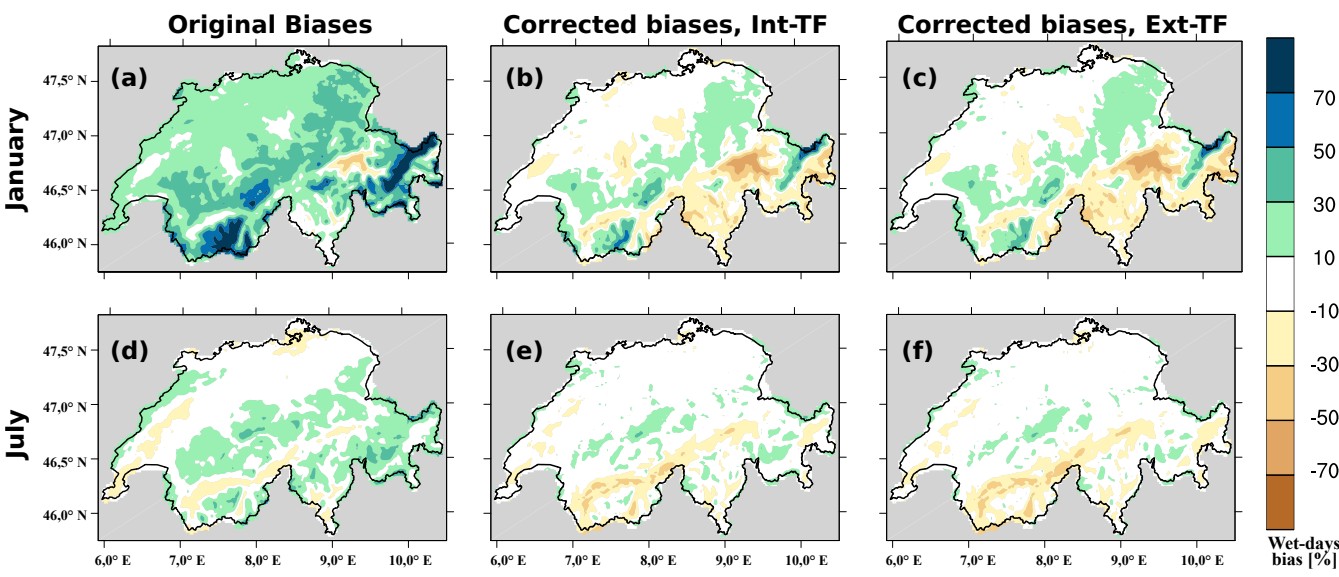

**Figure 9.** Biases in the wet-day frequency within the 30-years period over Switzerland. (a) represents the original biases in January, (b) the biases after being corrected using Int-TFs in January, (c) the biases after being corrected using Ext-TFs in January, (d), (e), and (f) as (a), (b), and (c) but in July, respectively.

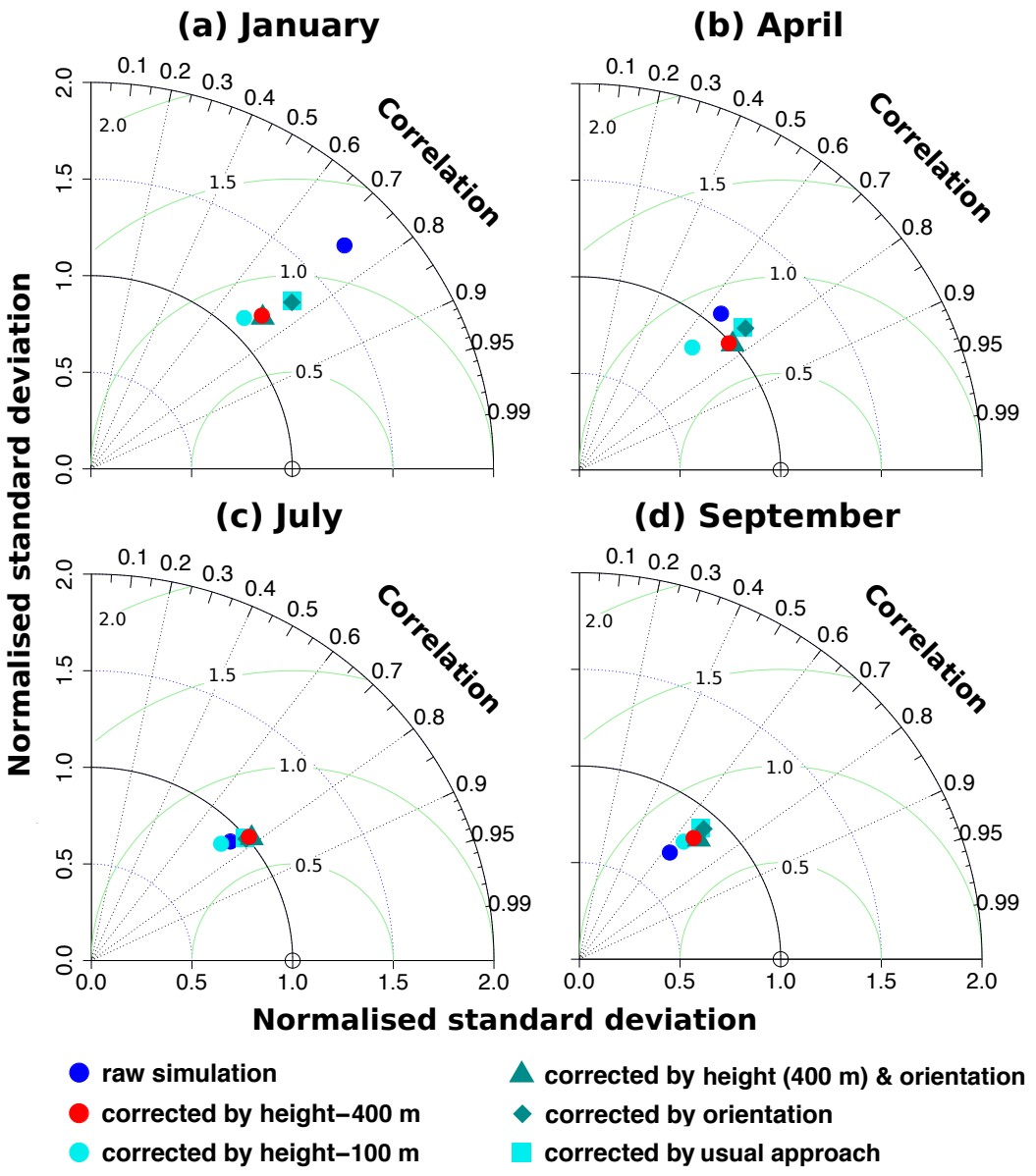

**Figure 10.** Performance of bias-correction with different settings. (a) shows a Taylor diagram for January, (b) for April, (c) for July and (d) for September. Blue dots represent the raw simulation, red dots the simulation corrected by using height-intervals of 400 m, cyan dots the simulation corrected by using height-intervals of 100 m, petrol triangles the simulation corrected by using height-intervals of 400 m and slope-orientations, petrol diamonds the simulation corrected by slope-orientations, and cyan squares the simulation corrected by the simple approach (the entire Swiss region). Note that in the Taylor diagram the spatial correlation, spatial root-mean-square-error and spatial standard deviation are shown.

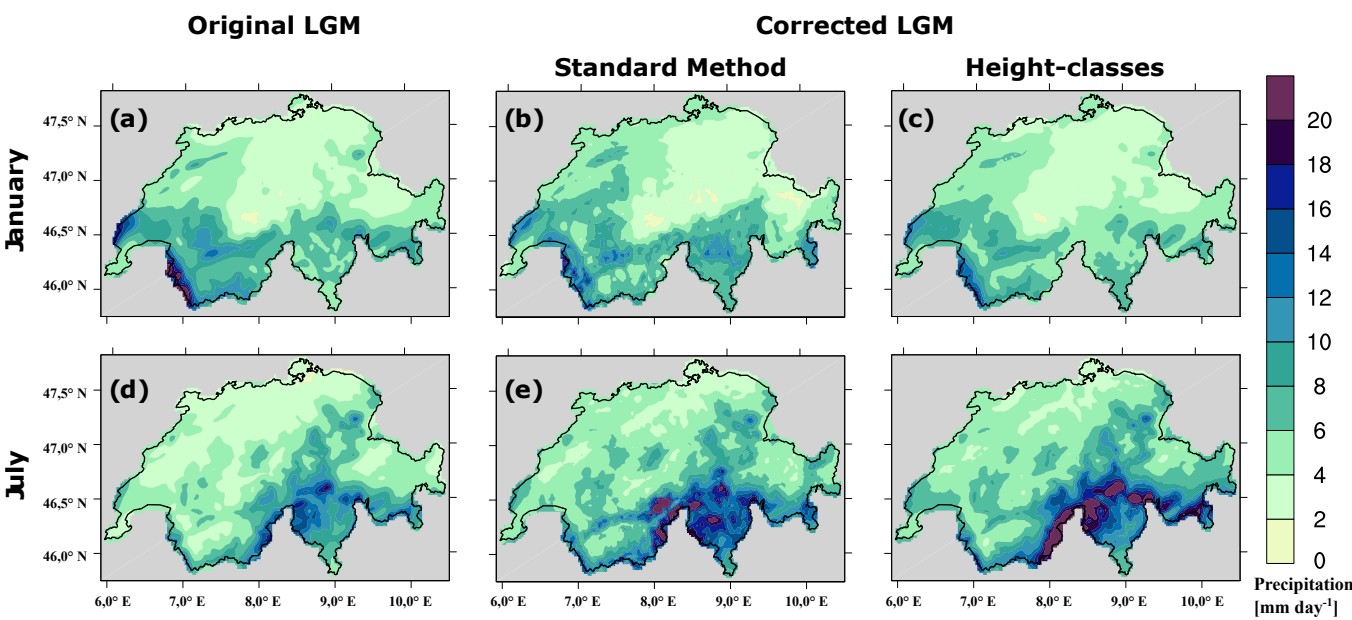

**Figure 11.** Monthly climatology of 30-years precipitation over Switzerland during the LGM. (a) represents uncorrected precipitation intensity in January, (b) as (a) but corrected using the standard EQM, (c) as (b) but using the new method, (d), (e) and (f) as (a), (b) and (c) but for July, respectively.

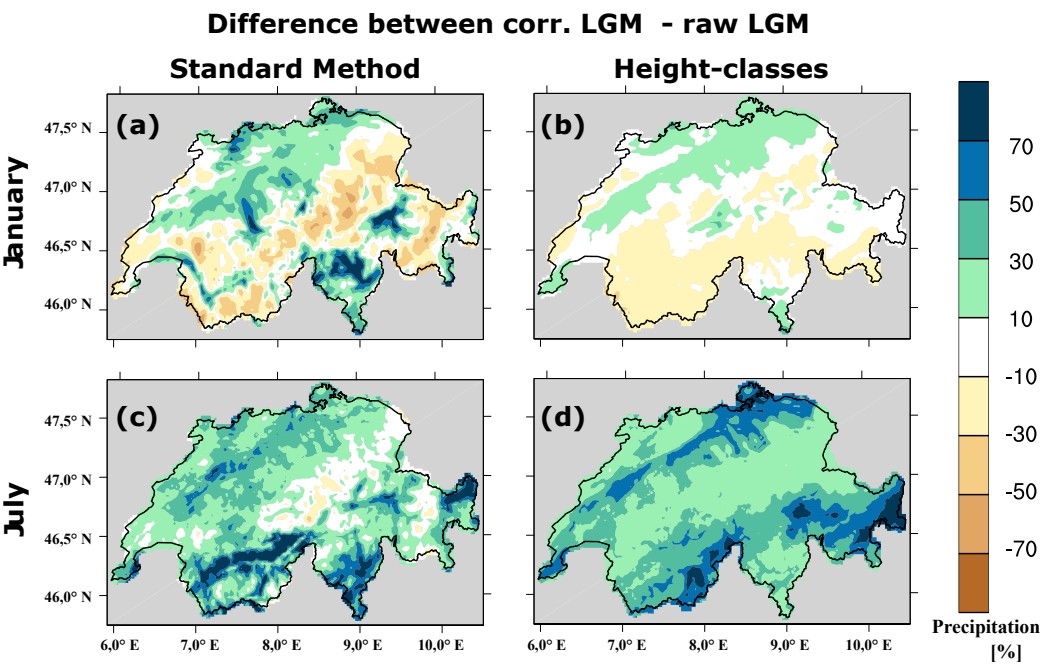

**Figure 12.** Performance of the correction for the monthly climatology of 30-years precipitation over Switzerland during the LGM. (a) represents the differences in January between corrected precipitation using the standard EQM and uncorrected precipitation, (b) as (a) but using the new method, (c) and (d) as (a) and b) but in July, respectively.

**Table 1.** External forcing used in Hofer et al. (2012a, b) for 1990 AD and LGM conditions.

| Parameter name | 1990 AD | LGM |
|---|---|---|
| TSI (W m$^{-2}$) | 1361.77 | 1360.89 |
| Eccentricity ($10^{-2}$) | 1.6708 | 1.8994 |
| Obliquity (°) | 23.441 | 22.949 |
| Angular precession (°) | 102.72 | 114.43 |
| CO2 (ppm) | 353.9 | 185 |
| CH4 (ppb) | 1693.6 | 350 |
| N2O (ppb) | 310.1 | 200 |

**Table 2.** Important parameterisations used to run WRF.

| Parameterisation | Parameter name | Chosen parameterisation | Applied to |
|---|---|---|---|
| Microphysics | mp_physics | WRF single moment 6-class scheme | Domain 1 – 4 |
| Longwave radiation | ra_lw_physiscs | RRTM scheme | Domain 1 – 4 |
| Shortwave radiation | ra_sw_physics | Dudhia scheme | Domain 1 – 4 |
| Surface layer | sf_sfclay_pysics | MM5 similarity | Domain 1 – 4 |
| Land/water surface | sf_surface_physics | Noah–Multiparameterization Land Surface Model | Domain 1 – 4 |
| Planetary boundary layer | bl_pbl_physics | Yonsei University scheme | Domain 1 – 4 |
| Cumulus | cu_physics | Kain–Fritsch scheme | Domain 1 – 2 |
| | | No parameterisation | Domain 3 – 4 |