# Peer review of "A new bias-correction method for precipitation over complex terrain suitable for different climate states: a case study using WRF (version 3.8.1)"

_Geoscientific Model Development, 2019_

## Referee Comment (RC1) · Anonymous Referee #1 · 24 Jul 2019

The manuscript of Velasquez et al introduces and validates a new method to correct systematic precipitation biases of a regional climate model simulation in the complex terrain of the Alps. The correction is largely based on the quantile mapping method but explicitly takes into account orographic characteristics such as elevation and slope aspect. The 30-year long climate simulation is driven by a global climate model simulation and several intermediate regional nests. Simulated precipitation amounts are corrected towards two gridded precipitation datasets covering Switzerland and the Greater Alpine Region, respectively. The model evaluation, in general, shows a satisfying performance.

[Figure]
The topic of the work fits well into the journal's scope and is, in principle, relevant for its readership. However, in my opinion the manuscript suffers from a number of severe shortcomings and misunderstandings. These definitely need to be improved before a publication can be recommended. I'm listing the major as well as a few minor points below. The mentioned issues need to be worked on, and my recommendation is therefore to return the manuscript to the authors for major revision. I hope my comments are helpful to further improve the work.

Major points

- The underlying assumption of the presented exercise is that orographically defined classes are informative for the model's precipitation bias. In my opinion, this has not yet been convincingly shown. What would be required, for instance, is an analysis of the range of model biases WITHIN the individual orographic classes. Do classes separate from each other in such an analysis? Figure 7 provides an indication that this is not the case, as the spatial correlation does not systematically improve after application of the bias correction.

- As stated by the authors, the rationale behind the newly developed method is that bias correction would be possible for paleo climatic states subject to a different land surface topography (Alpine ice shield, for instance). There is a considerable danger that applying a correction method that is trained in today's climate does not hold for such a climatic state even if orography is considered as a co-variate in the bias correction. Large scale flow conditions, for instance, could be strongly different from today's conditions leading to a completely different bias structure even for the same orography class. Also, in a much colder climate the relation of snowfall to liquid precipitation would increase which might, in turn, lead to completely different model biases even for the same orographic class. To show that the assumption is valid, one would have to go much further with the modelling exercise. One could, for instance, carry out a second simulation with the very same GCM forcing but a modified Alpine topography in the RCM, and then apply the bias correction calibrated in the standard simulation with true

orography. Would the bias-correction produce a realistic precipitation pattern in such a disturbed simulation?

- The introduction definitely needs to be worked on and be streamlined. It currently includes quite some repetition, and the line of argumentation is not always straight. Some basic references (for instance on the evaluation of CORDEX experiments in Europe and over the Alps) are missing.

- At several points in the paper the authors mention that the traditional QM approach would calibrate one correction function for the entire domain. This is certainly not true. In a pure bias correction setting (raw grid = target grid) a separate correction function is calibrated for each individual grid cell.

- The reason for the second bias correction step (first part of local intensity scaling) remains completely unclear to me. The third step (QM) would account for this already (by adjusting the percentiles).

- The general setup of the bias correction remains unclear. Is the correction carried out grid cell by grid cell, or in a bulk manner for each orographic class?

- Figure 3 is unclear. What do the boxplots represent and what is the true y-axis scale? Do the boxplots cover the spatial variability of monthly mean precipitation for the entire domain (a) or the elevation classes (b,c)? The text mentions that daily precipitation variability is shown, but how does this aggregate to monthly precipitation (y-axis label) then? If boxplots really show the distribution of daily precipitation values does it really make sense to use the IQR? Depending on the wet day frequency more than 25% of the days might be dry, for instance.

- Also the general validation setup remains unclear to some extent, the validation technique and the respective reference datasets used needs to be better described. It is sometimes unclear whether the Swiss 2 km serves as reference or the Alpine 5 km grid.

- Any kind of bias correction will only be as good and as appropriate as the observational reference. The validity of an analysis of elevation dependencies and slope dependencies at regional scales in the gridded observational precipitation datasets needs to be discussed. Does the reference grid really represent such dependencies?

- The application of the Ext-TFs mixes spatial scales (classes based on 5 km orography vs. classes based on 2 km orography). This is potentially dangerous and the effects of this mismatch should be shown. Why is the validation, in this case, not carried out on the 5 km scale as well?

Minor points

- page 1 line 19: "is" instead of "has been"

- page 2 line 20: What is meant by "weaker intensity" here? Unclear.

- page 2 lines 16-19: Line of argumentation unclear. RCMs were already referred to just above (line 12ff)

- page 4 lines 1-2: No true in general. Ban et al. for instance show that mean precipitation can also be much worse in convection resolving experiments. Certain aspects (such as the diurnal cycle) are improved, but not all.

- page 4 lines 4-7: I don't really understand the reason behind this splitting in ten single 3-year simulations. 2 months spin up is certainly not enough for soil parameters and snow. Some more information on the setup and on the rationale behind it needs to be provided.

- page 4 lines 19-20: I guess this is hardly true. In areas where no observations are available gridded products can be subject to very high uncertainties as inter- and extrapolation are required here.

- page 5 lines 4-9: It remains unclear how these classes are computed. Based on the relation of a grid cell to its 8 direct neighbor grid cells? Please clarify.

- page 5 lines 15-17: Which threshold is then used in the present work?

- page 7 lines 30-32: This explanation seems to be not very likely given the turnaround time of atmospheric water vapor (a couple of days only). Water vapor should also frequently be resupplied by the boundary forcing of the RCM. Can you back this up by some reference?

- Figure 1: Why are Italy and Slovenia excluded from the Ext-TF analysis? They are part of the APGD dataset.

- Figures 4 and 5: Sorry, but it is unclear to me which bias is shown in these two figures. Bias of the IQR of daily precipitation amount sin Figure 5? Which intensity in Figure 4? Mean wet day intensity? Needs to be better explained.

---

## Short Comment (SC1) · 26 Jul 2019

Dear authors,

in my role as Executive editor of GMD, I would like to bring to your attention our Editorial version 1.2:

https://www.geosci-model-dev.net/12/2215/2019/

This highlights some requirements of papers published in GMD, which is also available on the GMD website in the 'Manuscript Types' section:

http://www.geoscientific-model-development.net/submission/manuscript_types.html

In particular, please note that for your paper, the following requirements have not been met in the Discussions paper:

- "The main paper must give the model name and version number (or other unique identifier) in the title."

- "If the model development relates to a single model then the model name and the version number must be included in the title of the paper. If the main intention of an article is to make a general (i.e. model independent) statement about the usefulness of a new development, but the usefulness is shown with the help of one specific model, the model name and version number must be stated in the title. The title could have a form such as, "Title outlining amazing generic advance: a case study with Model XXX (version Y)"."

To fulfill these criteria the title of your manuscript needs to be expanded to something like: "A new bias-correction method for precipitation over complex terrain suitable for different climate states: a case study using WRF (version 3.8.1)"

Yours,

Astrid Kerkweg

––––––––––––––––––––––––––––

---

## Referee Comment (RC2) · Anonymous Referee #2 · 9 Sep 2019

Review of

'A new bias-correction method for precipitation over complex terrain suitable for different climate states'

by P. Velasquez, M. Messmer, and C. Raible

**Recommendation: major revisions**

This manuscript presents a bias correction method that is based on Empirical Quantile Mapping (EQM) with corrections depending on elevation and slope orientation. It is tested using a 2km-resolution RCM simulation for the Alps driven by a GCM. The simulations are 30 years long and represent perpetual 1990 conditions. The method is validated against precipitation observations for Switzerland, with the EQM fitted either using data for Switzerland, or a larger Alpine area. The intended application is bias correction of paleoclimate simulations, in which the topography over glaciated areas is different from current conditions.

The approach is novel, interesting and potentially useful. However the manuscript has substantial shortcomings with respect to explaining the conceptual basis and technical details. Many of my concerns are essentially identical to those already raised by the first reviewer. I will thus not list them all, and comment only where I would like to emphasize issues or add some detail. Major rewriting addressing these comments is required before the manuscript can be considered for publication.

**Specific comments**

1)
The setup for the EMQ is completely unclear. Standard EQM is local, i.e. it would apply a different correction for each location for which observations are available, in this case for each gridcell of the observational datasets. There is no explanation of how the corrections for the subclasses (elevation and slope) are obtained. Are the local corrections averaged, or is the precipitation averaged prior to fitting the EQM?

This is obviously a key aspect of the method and it is surprising that it is not explained.

The statement that standard bias correction methods do not include the effect of topography is wrong, as the observations, which are the basis for the fitting, do include these effects. What is presumably meant is that standard bias correction does not include these effects explicitly, which means it cannot be applied when the topography changes.

2)
As already pointed out by the first reviewer, determining joint bias corrections for the subclasses defined by topography and slope only makes sense if the local bias corrections within a class are more similar than those between the classes. This needs to be shown.

3)
The justification for the intended application is superficial and ignores key problems. In turn this means that the justification for the new approach itself is weak. As pointed out already by the first reviewer, many things in addition to the topography are different in a glacial climate, for instance the large-scale circulation or the moisture content. It is thus highly questionable whether applying a bias correction that is based on present climate, even if it explicitly accounts for topography, would yield meaningful results.

This problem is closely related to the distinction of different types of errors and to the issue of propagation of GCM errors through dynamical downscaling. There are a few statements in the paper that mention that discrepancies of RCM simulations and observations might be caused by the driving GCM. However there is no systematic discussion of what kind of errors bias correction could correct in a meaningful way. A discussion of these issues can be found for instance in

Maraun, et al., 2017: Towards process-informed bias correction of climate change simulations. *Nature Climate Change,* **7**(11), 764-773

Maraun and Widmann, 2018: Statistical downscaling and bias correction in climate research. Cambridge University Press, ISBN 1107066050

Eden, J.M., Widmann, M., Grawe, D, and Rast. S., 2012: Reassessing the skill of GCM-simulated precipitation. *J. Climate*, **25**(11), 3970-3984.

4)
The fact that EQM leads to correct distributions for the fitting data is trivially true by construction. The informative part of the validation of statistical models is related to the aspects that are not  trivially in agreement with observations For each aspect of the validation it should be discussed to what extent a good skill can be expected by construction. For instance, given the unclear setup for fitting and application of the bias correction, it is not clear what causes the differences between observed and corrected distributions in Fig.3, or the differences in Fig. 4 and Fig. 5.

Some problems related to the validation of bias correction methods are discussed in

Maraun, D. and M. Widmann, 2018, 'Cross-validation of bias-corrected climate simulations is misleading', *HESS*, 22(9), 4867-4873.

5)
It is not clear why the wet-day frequency is adjusted prior to the fitting of the EQM. If EQM is applied to the whole distribution including dry days, this adjustment is included in the EQM fitting. The justification might be linked to the unexplained details in the fitting setup.

6)
Although it is mentioned that the errors in the observations should be taken into account when interpreting  the results, there is no substantial effort to actually do this. For instance it would be instructive to do a rough correction for the substantial  undercatch of precipitation falling as snow, which strongly affects the high elevations, and assess to what extent  the validation results are sensitive to this error.

7)

As the realization of internal variability is different the observations and in a free-running GCM (as opposed to a reanalysis) some differences between observations and simulations will be due to internal variability. This effect should be roughly quantified, for instance by showing fitting and validating the method for 10 or 15 year sub-periods (which would lead to 9 or 4 possible combinations of fitting and validation subperiods).

---

## Author Comment (AC1) · 6 Oct 2019

**Response to Referee Comment**

We greatly thank the reviewer for the careful and thorough reading of this manuscript and for the thoughtful comments and constructive suggestions, which help to improve the quality of our manuscript. The comments have been carefully considered and responded. Please find below our response to each comment.

**Mayor points:**

1. The underlying assumption of the presented exercise is that orographically defined classes are informative for the model's precipitation bias. In my opinion, this has not yet been convincingly shown. What would be required, for instance, is an analysis of the range of model biases WITHIN the individual orographic classes. Do classes separate from each other in such an analysis? Figure 7 provides an indication that this is not the case, as the spatial correlation does not systematically improve after application of the bias correction.

**RESPONSE:**

We thank the reviewer for bringing to our attention that we missed to show clearly enough the orographic dependence of the biases. To clarify this, we have attached a figure that presents the monthly mean biases for each height-class before and after the correction (Fig. R1). Figure R1 illustrates an overestimation at high elevations and an underestimation at the lower ones during the colder months. Moreover, different levels of underestimation are observed across the height-classes during the warmer months. Thus, the splitting into different height-classes is appropriate to be used in the bias correction. Moreover, we would like to mention that we explicitly present the model biases within two classes in the Fig. 3 (of the manuscript), and implicitly for all the height classes in Fig. 4 and 5. Therefore, we have included in the revised manuscript a more balanced discussion of how our approach is removing the biases.

Furthermore, we agree that the spatial correlation is only weakly improved. However, we would like to highlight here that we do not only consider the spatial correlation to assess the performance of the different corrections, but we also include the spatial standard deviation and the spatial root-mean-square-error. The Taylor-diagram in Fig. 7 (of the manuscript) shows all three parameters and thus, provides wider criteria than just considering spatial correlation.

Figure R1. Mean bias over Switzerland for different height-classes.

2. As stated by the authors, the rationale behind the newly developed method is that bias correction would be possible for paleo climatic states subject to a different land surface topography (Alpine ice shield, for instance). There is a considerable danger that applying a correction method that is trained in today's climate does not hold for such a climatic state even if orography is considered as a co-variate in the bias correction. Large scale flow conditions, for instance, could be strongly different from today's conditions leading to a completely different bias structure even for the same orography class. Also, in a much colder climate the relation of snowfall to liquid precipitation would increase which might, in turn, lead to completely different model biases even for the same orographic class. To show that the assumption is valid, one would have to go much further with the modelling exercise. One could, for instance, carry out a second simulation with the very same GCM forcing but a modified Alpine topography in the RCM, and then apply the bias correction calibrated in the standard simulation with true orography. Would the bias-correction produce a realistic precipitation pattern in such a disturbed simulation?

**RESPONSE:**

We appreciate this comment and recognize that the manuscript might lead to misunderstandings about the application of our bias-correction method to other climate states. The danger of correcting biases in a simulated climate with a method that has been trained with a climate that does not correspond to the simulated climate is well-known in the statistical downscaling methods. These are likewise calibrated with today's climate and applied to past and future climate states. Many statistical downscaling and correction methods suffer basically from the assumption of stationary biases, which implies that their algorithms trained with today's climate are considered to be also valid for different climate states. Thus, our work aims at presenting a new bias-correction that attempts to decrease this danger substantially by using orographic features as additional variables for the correction. Moreover, precipitation biases are not only produced by initial and boundary conditions provided by the global climate models, but also by parametrisations, physical and numerical formulations that are described in both global and regional climate models. The main goal of the presented work is to correct wet or dry biases that stem from either from global or regional models or both. These biases can be produced by parametrisations and numerical formulations, but those that are mainly associated with orographic effects, namely, vertical motion and precipitation-related processes. To clarify this, we will present a broader discussion on the general shortcomings of bias correction methods in the introduction and the conclusion section in the revised manuscript.

Furthermore, we also believe that the relation of snowfall – liquid precipitation would change in a much colder climate. However, this relation plays a negligible role in our correction method because the observational dataset and the model output, which are used in this work, consider both solid and liquid precipitation together. To clarify these points, we have added the definition of the precipitation in the manuscript and how days without precipitation are treated.

- Additional text on page 4 line 17

...Note that all data sets consider daily precipitation as total precipitation, i.e., both solid and liquid precipitation, and convective and non-convective precipitation. Moreover, days without precipitation are treated as censored values when daily precipitation is equal to 0 mm day4, although in the case of observations this is equivalent to 0.1 mm day4 due to gauge precision ...

The suggested sensitivity simulation would provide several problems. First, the global simulation would have to be rerun with an adapted alpine topography, as a circulation change should be expected when the Alps are reduced and increased. If inconsistent boundary conditions are given to the regional model this might lead to further errors that cannot be corrected by the proposed correction method. Second, this correction cannot be validated as there are no observations for such a climate, so the same problem as for past and future climates remains. Thus, we have used a different Alpine region to calibrate the correction method, which is considered as a different climate state due to its different precipitation pattern compared to the one from Switzerland (Frei and Schär, 1998). In addition, the corrected results can be easily evaluated using the gridded Swiss observational dataset, which is not the case in the suggested sensitivity test because there is not any modified observational dataset that considers changes in the topography.

Still, we agree that the method should be evaluated in a different climate state but this is beyond the scope of this publication. An idea to validate the proposed correction may be to simulate e.g. Last Glacial Maximum conditions and compare them to proxy data like alpine ice sheet extent. Such a validation would include some collaboration with glacier modellers that are able to use raw and corrected precipitation to predict glacier extents. With such a method, the correction could be much better verified than with the method suggested by the reviewer.

Frei, C., and C. Schär. 1998. 'A Precipitation Climatology of the Alps from High-Resolution Rain-Gauge Observations'. International Journal of Climatology 18 (8): 873–900. https://doi.org/10.1002/(SICI)1097-0088(19980630)18:8<873::AID-JOC255>3.0.CO;2-9.

3. The introduction definitely needs to be worked on and be streamlined. It currently includes quite some repetition, and the line of argumentation is not always straight. Some basic references (for instance on the evaluation of CORDEX experiments in Europe and over the Alps) are missing.

**RESPONSE:**

We greatly thank you for bringing to our attention that the introduction needs to be worked on. An improved introduction will be presented in the revised manuscript avoiding repetitions. Regarding the basic references, we would like to clarify that we point out the CORDEX experiments twice in the manuscript. First, it was brought up on page 2 line 14 when linking the precipitation biases with regional climate simulations. Second, we cited the work of Casanueva et al. (2016) on page 11 line 5, which is about an approach of correcting precipitation biases from some EURO-CORDEX RCMs. They mainly focus on Spain and the Alpine region.

Nevertheless, we agree that the CORDEX experiments are not fully mentioned in the manuscript and that they could be better introduced. Thus, we have included them more explicitly in the next version of the manuscript. We also will include a broader discussion on the limitations of bias correction methods.

4. At several points in the paper the authors mention that the traditional QM approach would calibrate one correction function for the entire domain. This is certainly not true. In a pure bias correction setting (raw grid = target grid) a separate correction function is calibrated for each individual grid cell.

**RESPONSE**:**

We fully agree that this statement needs to be considered for a reformulation, although a pure bias correction setting as mentioned by the reviewer (separate correction function calibrated for each grid point) would be also a statistical downscaling. Still, we have rephrased "commonly used method" into "simple approach" at various places throughout the manuscript and deleted some citations as follows:

- Page 6 lines 4 – 6:

...To demonstrate the improvement of using the new method, we further compare it to a commonly used method that is carried out without orographic features and uses TFs deduced for the entire region of Switzerland (2 km) (similar to Berg et al., 2012; Maraun, 2013; Fang et al., 2015) ...

... To demonstrate the improvement of using the new method, we further compare it to a simple method that is carried out without orographic features and uses TFs deduced for the entire region of Switzerland  $(2 \text{ km}) \dots$

- Page 8 lines 5 – 7:

...We assess in the following, which of these characteristics are necessary to improve the simple approach of applying one EQM to the entire domain, often used in studies for present day and future climate change (e.g., Evans et al., 2017; Li et al., 2017; Ivanov et al., 2018) ...

... We assess in the following, which of these characteristics are necessary to improve a simple approach of applying one EQM to the entire domain, where orographic features are not considered ...

- Page 11 lines 11 – 12

...Clearly, the new method outperforms the standard method of applying one EQM transfer function deduced for the entire region of interest, which is commonly used (Berg et al., 2012; Maraun, 2013; Fang et al., 2015) ...

...Clearly, the new method outperforms the simple method of applying one EQM transfer function that is deduced for the entire region of interest and does not consider any orographic features ...

**5. The reason for the second bias correction step (first part of local intensity scaling) remains completely unclear to me. The third step (QM) would account for this already (by adjusting the percentiles).**

**RESPONSE:**

We agree that the reason for the local intensity scaling method may not be fully explained. To clarify this point, it is necessary to mention the similarities and differences in the treatment of the very low intensity values between two quantile mapping techniques, namely, the parametric quantile mapping (QM) and the empirical quantile mapping (EQM). Both techniques treat days without precipitation as censored values and consider only days with precipitation. The QM obtains the quantiles and transfer functions (TFs) from a cumulative distribution function (CDF) that is previously fitted, and thus it could properly handle the very low values with an adequate distribution fitting. Whereas in our study, an empirical CDF is used to directly calculate the quantiles and TFs, which is the core of the EQM. The reason of using an EQM is because we do not assume any known distribution either in our data sets or in the possible application to other climate states. However, the results of the EQM can become unrealistic if the very low intensity values are not adjusted previously. The reason for this is that these values can produce inappropriate TFs due to an important shift in the distribution, i.e., the quantiles.

To adjust these very low values, an additional parameter is included in the definition of days without precipitation that has been mentioned before in the response to the second major point. The days without precipitation are considered as censored values when they fall below a certain threshold. Many studies use a static threshold that is between 0.01 and 1.00 mm day4, whereas in our study, we calculate different thresholds to be consistent with the differentiate biasestreatment across the groups (or subgroups) and months of the year. The threshold is calculated using the local intensity scaling method and can vary between 0.001 and 1.00 mm day4.

Changes in the manuscript are presented as follows:

**- Page 5 lines 13 – 14**

...2010). To correct precipitation with very low-intensity the first part of the local intensity scaling method is used (Schmidli et al., 2006). It consists ...

...2010), which can distort the precipitation distribution substantially (Teutschbein and Seibert, 2012). To correct precipitation with very low intensity, an additional parameter is included in the definition of dry days related with the uncorrected precipitation. The dry days are now considered as censored values when they fall below a certain threshold. Many studies use a static threshold that is between 0.01 and 1.00 mm day4 (Piani et al., 2010a; Lafon et al., 2013;

Maraun, 2013), whereas in our study, we calculate different thresholds to be consistent with the different biases-treatment across the groups (or subgroups) and months of the year. Then, we carry out the first part of the local intensity scaling method (Schmidli et al., 2006) that is also used by Teutschbein and Seibert (2012) before using the quantile mapping technique. This method consists ...

**- Page 5 lines 16 – 17**

...The threshold can vary from group to group, but it is often close to or smaller than 1 mm day4 Schmidli et al., 2006).

...In our work, the threshold can vary from group to group and from month to month between 0.001 and 1 mm day-1 as in Schmidli et al. (2006) ...

**6. The general setup of the bias correction remains unclear. Is the correction carried out grid cell by grid cell, or in a bulk manner for each orographic class?**

**RESPONSE:**

We thank you for bringing to our attention that the general setup of the bias correction remains unclear. To clarify it we have changed lines 31 - 32 on page 5 as follows:

...To combine all steps, the EQM is applied to each (sub-) group and each month of the year, separately. This results in a set of TFs for each (sub-) group and each month of the year. Thus...

...To combine all steps, the local intensity scaling method and the EQM are applied to each (sub-) group defined in the first step and each month of the year, separately, by pooling all grid points that belong to it and handling them as a single distribution of daily precipitation. This results in a set of TFs for each (sub-) group and each month of the year. Moreover, the correction is afterwards applied to the daily precipitation in every grid point using the TFs that are common to all elements within the same group (or sub-group) and month. Thus...

7. Figure 3 is unclear. What do the boxplots represent and what is the true y-axis scale? Do the boxplots cover the spatial variability of monthly mean precipitation for the entire domain (a) or the elevation classes (b,c)? The text mentions that daily precipitation variability is shown, but how does this aggregate to monthly precipitation (y-axis label) then? If boxplots really show the distribution of daily precipitation values does it really make sense to use the IQR? Depending on the wet day frequency more than 25% of the days might be dry, for instance.

**RESPONSE**:**

We appreciate that you bring to our attention that the y-axis, the caption and the text are confusing. To clarify this, we would like to mention that the boxplots illustrate the spatial distribution of monthly mean values of precipitation intensity across a specific area within 30 years. Thus, we have modified them as follows:

**- The y-axis**

Monthly precipitation [mm day-]

**Intensity [mm day-]**

**- The text in the caption**

Boxplots are illustrating the annual cycle and monthly distribution of daily precipitation: (a) entire Switzerland, (b) all grid points in the height class of 400 - 800 m, and (c) of 2.800 - 3.200 m. Black box-plots represent the observations (RhiresD data), blue and red ones the raw and corrected simulation, respectively. Top and bottom ends of the dashed lines represent the maximum and minimum values, respectively. Dots represent the mean.

Boxplots illustrate the spatial distribution of monthly mean values of precipitation intensity across a specific area within 30 years: (a) the area covers all grid points over entire Switzerland, (b) the grid points in the height class of 400 - 800 m, and (c) the grid points in the height class of 2.800 - 3.200 m. Black box-plots represent the observations (RhiresD data), blue and red ones the raw and corrected simulation, respectively. Top and bottom ends of the dashed lines represent the maximum and minimum values, respectively. Dots represent the spatial climatological mean value.

**- Text, page 6 line 19 – 20**

..., the annual cycle and the monthly distributions of daily precipitation are ...

**..., the annual cycle and the distributions of monthly mean precipitation intensity are ...**

**- Text, page 6 line 32 – 33**

... For these example months, we present the patterns of biases in precipitation, changes in the distribution of daily precipitation, illustrated by the interquartile range as well as biases in wet-day frequency ...

... For these example months, we present the spatial patterns of the biases in the mean precipitation intensity, in the variability illustrated by the interquartile range, and in the wet-day frequency ...

**8. Also the general validation setup remains unclear to some extent, the validation technique and the respective reference datasets used needs to be better described. It is sometimes unclear whether the Swiss 2 km serves as reference or the Alpine 5 km grid.**

**RESPONSE**:**

We agree that the validation technique and the data sets used are not fully described. To clarify it, we have modified it as follows:

**- Page 5 lines 33 - 35 and page 6 lines 1 - 6**

...To come up with a final method for the Alpine region we first test the influence of the different orographic characteristics (step 1). To be consistent with former studies (e.g., Sun et al., 2011; Themessl et al., 2012; Wilcke et al., 2013; Rajczak et al., 2016), the evaluation of the new method first uses the same region where the TFs are estimated. To be more rigorous, we additionally apply a cross-validation: Thereby, Switzerland is defined as the area to be corrected; then, we calculate two different TFs; namely, from the same Swiss region called Internal TFs (Int-TF), and from the corresponding Alpine region of Germany, France, and Austria altogether called External TFs (Ext-TF) (Fig. 1c). Note that Ext-TFs are carried out at 5 km horizontal resolution. To demonstrate the improvement of using the new method, we further compare it to a commonly used method that is carried out without orographic features and uses TFs deduced for the entire region of Switzerland (2 km) (similar to Berg et al., 2012; Maraun, 2013; Fang et al., 2015) ...

... To come up with a final method for the Alpine region we first evaluate the influence of the different orographic characteristics (step 1). To be consistent with former studies (e.g., Sun et al., 2011; Themessl et al., 2012; Wilcke et al., 2013; Rajczak et al., 2016), the evaluation uses the same region where the TFs are estimated. Explicitly, this means that the Swiss region in the WRF output (2 km) is defined as the area to be corrected and the RhiresD data set (at 2 km resolution) is used to obtain the TFs and to evaluate the different correction methods. Once the final method is determined, we additionally apply a cross-validation to test the method more rigorously: Thereby, Switzerland is defined as the area to be corrected (WRF output at 2 km resolution); then, we calculate two sets of TFs. The first one is obtained from the same Swiss region called Internal TFs (Int-TF) using the RhiresD data set (at 2 km resolution), and the second one from the corresponding Alpine region of Germany, France, and Austria altogether called External TFs (Ext-TF) using the APGD data set (at 5 km resolution; Fig. 1c). Note that Ext-TFs are carried out at 5 km horizontal resolution and applied to Switzerland at 2 km resolution. To demonstrate the improvement of using the new method, we further compare it to a simple method that is carried out without orographic features and uses TFs deduced for the entire region of Switzerland (at 2 km resolution) ...

9. Any kind of bias correction will only be as good and as appropriate as the observational reference. The validity of an analysis of elevation dependencies and slope dependencies at regional scales in the gridded observational precipitation datasets needs to be discussed. Does the reference grid really represent such dependencies?

**RESPONSE**:**

We appreciate this comment. We agree that we missed to show the validity of the elevation and slope dependencies in the gridded observational data sets. Note that the observational data sets have a height dependence on its quality. To clarify this, a discussion will be presented in the revised manuscript.

Still, the observational data sets are considered generally reliable and represent orographic features well, although at high altitudes less data sets are available (Fig. R2; Isotta et al. 2014). Note that in this study we do not explicitly consider any uncertainty, and instead assume that these observations represent the true precipitation without errors. Still, we will discuss the uncertainty issue in particular for the results in high altitudes.

---

## Author Comment (AC2) · 6 Oct 2019

Dear Dr. Astrid Kerkweg as Executive editor,

We greatly appreciate your comment. The new title will be presented in the revised manuscript.

Best regards,

On behalf of co-authors,

Patricio Velasquez

---

## Author Comment (AC3) · 6 Oct 2019

**Response to Referee Comment**

We appreciate the time taken by the reviewer for the careful and thorough reading of this manuscript and for considering the remarks of the first reviewer. The additional clarifications and constructive suggestions will certainly help to improve the quality of our manuscript. The comments have been carefully considered and responded. Please find our response to each comment below.

**Specific comments:**

**1.** *The setup for the EMQ is completely unclear. Standard EQM is local, i.e. it would apply a different correction for each location for which observations are available, in this case for each gridcell of the observational datasets. There is no explanation of how the corrections for the subclasses (elevation and slope) are obtained. Are the local corrections averaged, or is the precipitation averaged prior to fitting the EQM?*
*This is obviously a key aspect of the method and it is surprising that it is not explained.*
*The statement that standard bias correction methods do not include the effect of topography is wrong, as the observations, which are the basis for the fitting, do include these effects. What is presumably meant is that standard bias correction does not include these effects explicitly, which means it cannot be applied when the topography changes*

**RESPONSE**:

We agree that the setup of the bias correction remains unclear. Still we would like to point out that one strength of our method is that it is not local (the standard EQM described by the reviewer is a bias correction plus statistical downscaling). The simple reason is that a localized correction would fail in different states like the Last Glacial Maximum as valleys are filled with ice. To make the suggested method clearer, we have modified the manuscript as follows:

- Page 5 lines 31 – 32

…To combine all steps, the EQM is applied to each (sub-) group and each month of the year, separately. This results in a set of TFs for each (sub-) group and each month of the year. Thus…

…To combine all steps, the local intensity scaling method and the EQM are applied to each (sub-) group defined in the first step and to each month of the year, separately, by pooling all grid points that belong to each group and handling them as a single distribution of daily precipitation. This results in a set of TFs for each (sub-) group and each month of the year. For instance, when the correction is carried out using height-classes of 400 m, a TF is defined for each group, resulting in nine TFs for each month and in total 108 TFs throughout the year. Moreover, the correction is afterwards applied to the daily precipitation in every grid point using the TFs that are common to all elements within the same group (or sub-group) and month. Thus…

We also agree that the observational data sets implicitly include effects of topography. Changes regarding this point are presented in the following lines of the manuscript:

- Page 2 line 33

…correction methods do not consider orographic features that…

…correction methods do only implicitly consider orographic features that…

- Page 3 line 6

…time includes orographic characteristics…

…time explicitly includes orographic characteristics…

**2.** ***As already pointed out by the first reviewer, determining joint bias corrections for the subclasses defined by topography and slope only makes sense if the local bias corrections within a class are more similar than those between the classes. This needs to be shown***

**RESPONSE**:

We appreciate this comment and we agree that we missed to show clearly enough the argumentation for using different classes. As reviewer 1 asked a similar question we present there the same answer: We thank the reviewer for bringing to our attention that we missed to show clearly enough the orographic dependence of the biases. To clarify this, we have attached a figure that presents the monthly mean biases for each height-class before and after the correction (Fig. R1). Figure R1 illustrates an overestimation at high elevations and an underestimation at the lower ones during the colder months. Moreover, different levels of underestimation are observed across the height-classes during the warmer months. Thus, the splitting into different height-classes is appropriate to be used in the bias correction. Moreover, we would like to mention that we explicitly present the model biases within two classes in the Fig. 3 (of the manuscript), and implicitly for all the height classes in Fig. 4 and 5. Note that the biases within the classes are much smaller than between the classes. Therefore, we have included in the revised manuscript a more balanced discussion of how our approach is removing the biases.

[Figure]

Figure R1. Mean bias over Switzerland for different height-classes.

**3.**    *The justification for the intended application is superficial and ignores key problems. In turn this means that the justification for the new approach itself is weak. As pointed out already by the first reviewer, many things in addition to the topography are different in a glacial climate, for instance the large-scale circulation or the moisture content. It is thus highly questionable whether applying a bias correction that is based on present climate, even if it explicitly accounts for topography, would yield meaningful results.*
*This problem is closely related to the distinction of different types of errors and to the issue of propagation of GCM errors through dynamical downscaling. There are a few statements in the paper that mention that discrepancies of RCM simulations and observations might be caused by the driving GCM. However, there is no systematic discussion of what kind of errors bias correction could correct in a meaningful way. A discussion of these issues can be found for instance in*
*Maraun, et al., 2017: Towards process-informed bias correction of climate change simulations. Nature Climate Change, 7(11), 764-773*
*Maraun and Widmann, 2018: Statistical downscaling and bias correction in climate research. Cambridge University Press, ISBN 1107066050*
*Eden, J.M., Widmann, M., Grawe, D, and Rast. S., 2012: Reassessing the skill of GCM-simulated precipitation. J. Climate, 25(11), 3970-3984.*

**RESPONSE**:

We appreciate that the reviewer brings up the point that it might be misleading about to what extent the presented bias-correction can be applied to other climate states. As already responded to reviewer 1, we would like to mention that the danger of correcting biases in a simulated climate with a method that has been trained with a climate that does not correspond to the simulated one is well-known in the statistical downscaling and correction methods. Statistical downscaling and correction methods suffer basically from the assumption of stationary biases, which implies that their algorithms trained with today's climate are considered to be also valid for different climate states. Thus, our work aims at presenting a new bias-correction that attempts to decrease this danger substantially by using orographic features as additional variables for the correction. Note, that the presented correction is obviously only applicable in regions where the topography is rather complex and where topography has certainly an influence on the local atmospheric circulation.

Moreover, precipitation biases are not only produced by initial and boundary conditions provided by the global climate models, but also by parametrisations, physical and numerical formulations that are described in both global and regional climate models. The main goal of the presented work is to correct wet or dry biases that come either from global or regional models or both. These biases can be produced by parametrisations and numerical formulations, but mainly by those that are associated with orography effects, namely, vertical motion and precipitation-related processes over complex terrain. To clarify this, we will present a discussion of the abilities and limitations of the presented correction in the results part of the revised manuscript. As responded to reviewer 1 we also include some general discussion on bias correction methods in the introduction part including the references suggested.

**4.**    *The fact that EQM leads to correct distributions for the fitting data is trivially true by construction. The informative part of the validation of statistical models is related to the aspects that are not trivially in agreement with observations. For each aspect of the validation it should be discussed to what extent a good skill can be expected by construction.*

*For instance, given the unclear setup for fitting and application of the bias correction, it is not clear what causes the differences between observed and corrected distributions in Fig.3, or the differences in Fig. 4 and Fig. 5.*
*Some problems related to the validation of bias correction methods are discussed in Maraun, D. and M. Widmann, 2018, 'Cross-validation of bias-corrected climate simulations is misleading', HESS, 22(9), 4867-4873.*

**RESPONSE**:

We thank the reviewer for this comment and agree that the validation may be poorly discussed. As noted by Bennett et al. (2014), the importance of cross-validation methods is that they can test the ability of bias-correction techniques on a different climate state. However, this might not be reasonable as the biases of the other climate state may not remain unchanged and the method's accomplishment relies on the biases caught during the period the method is trained on. We also recognise that recent studies by Maraun et al. (2017) and Maraun and Widmann (2018) have argued against carrying out a cross-validation for evaluating bias corrections. The authors remark that the observational and simulated data sets do not have a synchronised internal climate variability. Thus, this asynchronism in the internal climate variability may be one of the sources of the biases in free-running models.

Furthermore, as mentioned by Maraun and Widmann (2018), our cross-validation method does not compare the correction to the observations on the validation period (future or past climate state), which can produce false positive or true negative results due to internal variability in the model or observations, but the method assesses whether the statistical evolution of the model is kept.

Moreover, one of the reasons that may explain the remaining difference between the observational and the corrected data sets, as mentioned in the manuscript, can be traced back to the fact that some height classes sample over regions with slightly different biases. Hence, biases of one area can be diminished by the biases that are shared by the other areas. For instance, the strong negative biases observed in the Rhone Valley and Ticino are not fully corrected because the slight underestimation across the Swiss Plateau dominates the bias in this height-class.

Nevertheless, we agree that the evaluation and the argumentation for the remaining biases is not discussed clearly enough in the manuscript and that this should be better explained. Thus, we will include such an evaluation more explicitly in the results part of next version of the manuscript.

Bennett, James C., Michael R. Grose, Stuart P. Corney, Christopher J. White, Gregory K. Holz, Jack J. Katzfey, David A. Post, and Nathaniel L. Bindoff. 2014. 'Performance of an empirical bias-correction of a high-resolution climate dataset'. International Journal of Climatology 34 (7): 2189–2204. https://doi.org/10.1002/joc.3830.

**5.** *It is not clear why the wet-day frequency is adjusted prior to the fitting of the EQM. If EQM is applied to the whole distribution including dry days, this adjustment is included in the EQM fitting. The justification might be linked to the unexplained details in the fitting setup.*

**RESPONSE**:

We thank the reviewer for highlighting this point and recognize that this adjustment may not be clear enough. We would like to mention that the adjustment does not mainly focus on the wet-day frequency, but the very low intensity values. As clarified already in the answer for reviewer 1, we agree that the argumentation for this adjustment may not be fully explained. To make this clear, we would like to mention that, in our study, we use an empirical quantile mapping technique (EQM) that differs from the parametric quantile mapping technique (QM). The reason of using an EQM is because this technique uses an empirical cumulative distribution function and does not fit any parametric distribution to the sample, i.e, (sub-) groups, as it is done in the QM. Therefore, we do not assume any known distribution either in our data sets or in the possible application to other climate states. However, the results of the EQM can become unrealistic if the very low intensity values are not adjusted previously. The reason for this is that the very small values can produce inappropriate TFs due to an important shift in the distribution, i.e., the quantiles.

To adjust these very low values, an additional parameter is included in the definition of days without precipitation that has been mentioned in the respond of the second major point of reviewer 1. The days without precipitation are now considered as censored values when they fall below a certain threshold. Many studies use a static threshold that is between 0.01 and 1.00 mm day$^{-1}$, whereas in our study, we calculate different thresholds to be consistent with the differentiate biases-treatment across the groups (or subgroups) and months of the year. The threshold is calculated using the local intensity scaling method and can vary between 0.001 and 1.00 mm day$^{-1}$. To clarify this, we have made some changes that are presented in the revised manuscript and also in response to the fifth major comment of reviewer 1.

**6.** ***Although it is mentioned that the errors in the observations should be taken into account when interpreting the results, there is no substantial effort to actually do this. For instance, it would be instructive to do a rough correction for the substantial undercatch of precipitation falling as snow, which strongly affects the high elevations, and assess to what extent the validation results are sensitive to this error.***

**RESPONSE**:

We appreciate this comment. We agree that we missed to show a wider discussion about the error in the observational data sets when interpreting the results of the correction method. As mentioned by (Isotta, 2014), the gridded observational data sets do not only present errors due to the interpolation methods, but they also show errors that may differ in quantity from one to other station (Sevruk, 1985; Richter, 1995) and are related to the "gauge undercatch", whose magnitudes range from 5% over the flatland regions to 30% above 1500 m a.s.l.. Therefore, we will include a better discussion of these errors when analysing the correction, which will be presented in the results part of the revised manuscript.

Sevruk B. 1985. Systematischer Niederschlagmessfehler in der Schweiz. Der Niederschlag in der Schweiz, Beiträge zur. Geologischen Karte der Schweiz-Hydrologie 31: 65–75.

Richter D. 1995. Ergebnisse methodischer Untersuchungen zur Korrektur des systematischen Messfehlers des Hellmann-Niederschlagsmessers. Bericht Deutschen Wetterdienstes 194, 93 pp. (To be obtained from German Weather Service, Offenbach a.M., Germany.)

**7.** *As the realization of internal variability is different the observations and in a free-running GCM (as opposed to a reanalysis) some differences between observations and simulations will be due to internal variability. This effect should be roughly quantified, for instance by showing fitting and validating the method for 10 or 15 year sub-periods (which would lead to 9 or 4 possible combinations of fitting and validation subperiods).*

**RESPONSE**:

We thank the reviewer for bringing to our attention the approach to quantify the biases that may be caused by differences between the internal variability of the observational data set and the simulated one. Furthermore, we would like to mention that correction methods are sensitive to the period the methods are trained on, and their accuracies would increase as more information from the observational data sets is taken into account (Lafon et al., 2013). Therefore, since the accuracy of our correction method needs to be kept as high as possible, we will carry out the suggestion made by the reviewer by splitting the data sets into two sub-periods. The outcome will be presented in results and conclusion part of the revised manuscript.

Once again, we would like to thank the reviewer for the time invested to review our paper so carefully and we are looking forward to meeting the expectations.

Best regards,
On behalf of the co-authors,

Patricio Velasquez

---

## Author Response (AR1)

**Final Response to Referees**

We greatly thank the reviewers for the careful and thorough reading of our manuscript. The additional clarifications and constructive suggestions have certainly helped to improve the quality of our manuscript. The comments have been carefully considered and responded. Please find below our response to each comment.

**Response to Referee #1**

**Mayor points:**

*1.      The underlying assumption of the presented exercise is that orographically defined classes are informative for the model's precipitation bias. In my opinion, this has not yet been convincingly shown. What would be required, for instance, is an analysis of the range of model biases WITHIN the individual orographic classes. Do classes separate from each other in such an analysis? Figure 7 provides an indication that this is not the case, as the spatial correlation does not systematically improve after application of the bias correction.*

**RESPONSE**:

We thank the reviewer for bringing up this concern. We agree that the main purpose of the correction method might still be a bit unclear and we would like to clarify this in more details in the following. With the present study, we would like to obtain a flexible correction that can be applied to several different climate states at the same time. To obtain this, the correction method should not be constrained to the actual climate too much, this is, because circulation changes and atmospheric characteristics may be variable between different climates. We agree that cluster analysis of precipitation and its errors should be applied, so that errors can be grouped accordingly and to keep the error within classes as small as possible, to obtain an optimal correction result. This has for example been performed by Gomez et al. (2018) for Switzerland. The drawback of such a correction for our purpose is that such a cluster analysis is always based on the characteristics and circulation of the current climate and this is what we would like to avoid as much as possible. To be as much independent from current climates as possible and to still provide a correction that still touches upon important characteristics in the Alpine climate, we came up with "static" characteristics, i.e. topography height and orientation. Both, topography and orientation will remain similar during different climate states, even if we are aware of the fact that in any correction the effect of topography is implicitly included. Nevertheless, we would like to show here that biases have some orographic dependence. To clarify this, we have attached a figure that presents the monthly mean biases for each height-class before and after the correction (Fig. R1). Figure R1 illustrates an overestimation at high elevations and an underestimation at the lower ones during the colder months. Moreover, different levels of underestimation are observed across the height-classes during the warmer months. Thus, the splitting into different height-classes is appropriate to be used in the bias correction. Moreover, we would like to mention that we explicitly present the model biases within two classes in Fig. 3 (of the manuscript), and implicitly for all the height classes in Fig. 4 and 5. Therefore, we have included a more balanced discussion about our approach in the results part of the revised manuscript.

Furthermore, we agree that the spatial correlation is only weakly improved. However, we would like to highlight here that we do not only consider the spatial correlation to assess the performance of the different corrections, but we also include the spatial standard deviation and the spatial root-mean-square-error. The Taylor-diagram in Fig. 7 (of the manuscript) shows all three parameters and thus, provides wider criteria than just considering spatial correlation.

[Figure]

Figure R1. Mean bias over Switzerland for different height-classes.

*2.     As stated by the authors, the rationale behind the newly developed method is that bias correction would be possible for paleo climatic states subject to a different land surface topography (Alpine ice shield, for instance). There is a considerable danger that applying a correction method that is trained in today's climate does not hold for such a climatic state even if orography is considered as a co-variate in the bias correction. Large scale flow conditions, for instance, could be strongly different from today's conditions leading to a completely different bias structure even for the same orography class. Also, in a much colder climate the relation of snowfall to liquid precipitation would increase which might, in turn, lead to completely different model biases even for the same orographic class. To show that the assumption is valid, one would have to go much further with the modelling exercise. One could, for instance, carry out a second simulation with the very same GCM forcing but a modified Alpine topography in the RCM, and then apply the bias correction calibrated in the standard simulation with true orography. Would the bias-correction produce a realistic precipitation pattern in such a disturbed simulation?*

**RESPONSE**:

We appreciate this comment and recognize that the manuscript might lead to misunderstandings about the application of our bias-correction method to other climate states. The danger of correcting biases in a simulated climate with a method that has been trained with a climate that does not correspond to the simulated climate is well-known in statistical downscaling methods. These are likewise calibrated with today's climate and applied to past and future climate states. Many statistical downscaling and correction methods suffer basically from the assumption of stationary biases, which implies that their algorithms trained with today's climate are considered to be also valid for different climate states. Thus, our work aims at presenting a new bias-correction that attempts to decrease this danger by using orographic

features, which are less likely characteristics of the current climate only. Moreover, precipitation biases are not only produced by initial and boundary conditions provided by the global climate models, but also by parametrisations, physical and numerical formulations that are described in both global and regional climate models. The main goal of the presented work is to correct wet or dry biases that stem either from global or regional models or both. These biases can be produced by parametrisations and numerical formulations, but those that are mainly associated with orographic effects, namely, vertical motion leading to precipitation. To clarify this, we extended the discussion on the general shortcomings of bias correction methods in the introduction and the conclusion section in the revised manuscript.

Furthermore, we agree that the relation of snowfall – liquid precipitation would change in a much colder climate. However, this relation plays a negligible role in our correction method because the observational dataset and the model output, which are used in this work, consider both solid and liquid precipitation together. To clarify these points, we have included the definition of the precipitation and the days without precipitation in the manuscript as follows.

- Additional text on page 4 line 17

…Note that all data sets consider daily precipitation as total precipitation, i.e., both solid and liquid precipitation, and convective and non-convective precipitation. Moreover, days without precipitation are treated as censored values, i.e., not considered in analysis, when daily precipitation is equal to 0 mm day$^{-1}$, although in the case of observations this is equivalent to 0.1 mm day$^{-1}$ due to gauge precision …

The suggested sensitivity simulation would provide several problems. First, the global simulation would have to be rerun with an adapted alpine topography, as a circulation change should be expected when the Alps are reduced and increased. If inconsistent boundary conditions are given to the regional model this might lead to further errors that cannot be corrected by the proposed correction method. Second, this correction cannot be validated as there are no observations for such a climate, so the same problem as for past and future climates remains. Thus, we have used a different Alpine region to calibrate the correction method, which is considered as a different climate state due to its different precipitation pattern compared to the one from Switzerland (Frei and Schär, 1998). In addition, the corrected results can be easily evaluated using the gridded Swiss observational dataset, which is not the case in the suggested sensitivity.

Still, we agree that the method should be evaluated in a different climate state but this is beyond the scope of this publication. An idea to validate the proposed correction may be to simulate e.g. Last Glacial Maximum conditions and compare them to proxy data like alpine ice sheet extent. Such a validation would include some collaboration with glacier modellers that are able to use raw and corrected precipitation to predict glacier extents. We think that such a method could provide a good way to verify the presented method as proxies could mimic the missing observations during glaciated times.

Frei, C., and C. Schär. 1998. 'A Precipitation Climatology of the Alps from High-Resolution Rain-Gauge Observations'. International Journal of Climatology 18 (8): 873–900. https://doi.org/10.1002/(SICI)1097-0088(19980630)18:8<873::AID-JOC255>3.0.CO;2-9.

***3.      The introduction definitely needs to be worked on and be streamlined. It currently includes quite some repetition, and the line of argumentation is not always straight. Some basic references (for instance on the evaluation of CORDEX experiments in Europe and over the Alps) are missing.***

**RESPONSE**:

We greatly thank you for bringing to our attention that the introduction needs to be worked on. An improved introduction is presented in the revised manuscript avoiding repetitions.
Regarding the basic references, we would like to clarify that we point out the CORDEX experiments twice in the manuscript. First, it was brought up on page 2 line 14 when linking the precipitation biases with regional climate simulations. Second, we cited the work of Casanueva et al. (2016) on page 11 line 5, which is about an approach of correcting precipitation biases from some EURO-CORDEX RCMs. They mainly focus on Spain and the Alpine region.
Nevertheless, we agree that the CORDEX experiments are not fully mentioned in the manuscript and that they could be better introduced. Thus, we have included them more explicitly in the next version of the manuscript.

***4.      At several points in the paper the authors mention that the traditional QM approach would calibrate one correction function for the entire domain. This is certainly not true. In a pure bias correction setting (raw grid = target grid) a separate correction function is calibrated for each individual grid cell.***

**RESPONSE**:

We fully agree that this statement needs to be considered for a reformulation, although a pure bias correction setting as mentioned by the reviewer (separate correction function calibrated for each grid point) would be also a statistical downscaling. Still, we have rephrased "commonly used method" into "simple approach" at various places throughout the manuscript and deleted some citations as follows:

- Page 6 lines 4 – 6:

…To demonstrate the improvement of using the new method, we further compare it to a commonly used method that is carried out without orographic features and uses TFs deduced for the entire region of Switzerland (2 km) (similar to Berg et al., 2012; Maraun, 2013; Fang et al., 2015) …

…To demonstrate the improvement of using the new method, we further compare it to a simple method that is carried out without orographic features and uses TFs deduced for the entire region of Switzerland (at 2 km resolution, 12 TFs in total) …

- Page 8 lines 5 – 7:

…We assess in the following, which of these characteristics are necessary to improve the simple approach of applying one EQM to the entire domain, often used in studies for present day and future climate change (e.g., Evans et al., 2017; Li et al., 2017; Ivanov et al., 2018) …

… We assess in the following, which of these characteristics are necessary to improve a simple approach of applying one EQM to the entire domain, where orographic features are not considered …

- Page 11 lines 11 – 12

…Clearly, the new method outperforms the standard method of applying one EQM transfer function deduced for the entire region of interest, which is commonly used (Berg et al., 2012; Maraun, 2013; Fang et al., 2015) …

…Clearly, the new method outperforms the simple method of applying one EQM transfer function that is deduced for the entire region of interest and does not consider any orographic features …

**5.** ***The reason for the second bias correction step (first part of local intensity scaling) remains completely unclear to me. The third step (QM) would account for this already (by adjusting the percentiles).***

RESPONSE:

We agree that the reason for the local intensity scaling method was not fully explained. To clarify this point, it is necessary to mention the similarities and differences in the treatment of the very low intensity values between two quantile mapping techniques, namely, the parametric quantile mapping (QM) and the empirical quantile mapping (EQM). Both techniques treat days without precipitation as censored values and consider only days with precipitation. The QM obtains the quantiles and transfer functions (TFs) from a cumulative distribution function (CDF) that is previously fitted, and thus it could properly handle the very low values with an adequate distribution fitting. Whereas in our study, an empirical CDF is used to directly calculate the quantiles and TFs, which is the core of the EQM. The reason of using an EQM is because we do not assume any known distribution either in our data sets or in the possible application to other climate states. However, the results of the EQM can become unrealistic if the very low intensity values are not adjusted previously. The reason for this is that these values can produce inappropriate TFs due to an important shift in the distribution, i.e., the quantiles (Teutschbein and Seibert, 2012; Lafon et al. 2013).

To adjust these very low values, an additional parameter is included in the definition of days without precipitation that has been mentioned before in the response to the second major point. The days without precipitation are not considered for calculating the TFs when they fall below a certain threshold. Many studies use a static threshold for the entire data set which is between 0.01 and 1.00 mm day$^{-1}$, whereas in our study, we calculate a static threshold for each group (or subgroup) and months of the year. This allows to be the consistent with the different biases-treatment across the groups (or subgroups) and months of the year. The threshold is calculated using the local intensity scaling method and can vary in our study from 0.001 to 1.00 mm day

Changes in the manuscript are presented as follows:

- Page 5 lines 13 – 14

…2010). To correct precipitation with very low-intensity the first part of the local intensity scaling method is used (Schmidli et al., 2006). It consists …

…2010), which can distort the precipitation distribution substantially, i.e., shifting the quantiles, producing inappropriate corrections in the third step when EQM is applied (Teutschbein and Seibert, 2012; Lafon et al., 2013). To correct precipitation with very low intensity, an additional parameter is included in the definition of dry days related with the uncorrected precipitation that is described in the section of model and data before. Dry days are not considered for calculating the TFs when they fall below a certain threshold. Many studies use a static threshold for the entire data set which is between 0.01 and 1.00 mm day[-1] (Piani et al., 2010a; Lafon et al., 2013; Maraun, 2013). We calculate a static threshold for each group (or subgroup) and months of the year. This allows to be the consistent with the different biases-treatment across the groups (or subgroups) and months of the year. Then, we carry out the local intensity scaling method (Schmidli et al., 2006) that is also used by Teutschbein and Seibert (2012) before using the quantile mapping technique. This method consists …

- Page 5 lines 16 – 17

…The threshold can vary from group to group, but it is often close to or smaller than 1 mm day[-1] Schmidli et al., 2006).

…In our work, the threshold can vary from group to group and from month to month between 0.001 and 1 mm day[-1], similar to Schmidli et al. (2006) …

**6.      The general setup of the bias correction remains unclear. Is the correction carried out grid cell by grid cell, or in a bulk manner for each orographic class?**

**RESPONSE:**

We thank the reviewer for bringing to our attention that the general setup of the bias correction remains unclear. To clarify it we have changed lines 31 – 32 on page 5 as follows:

…To combine all steps, the EQM is applied to each (sub-) group and each month of the year, separately. This results in a set of TFs for each (sub-) group and each month of the year. Thus…

…To combine all steps, the local intensity scaling method and the EQM are applied to each (sub-) group defined in the first step and each month of the year, separately, by pooling all grid points that belong to it and handling them as a single distribution of daily precipitation. This results in a set of TFs for each (sub-) group and each month of the year. For instance, when the correction is carried out using height-classes of 400 m, a TF is defined for each height group, resulting in nine TFs for each month and in total 108 TFs throughout the year. Moreover, the correction is afterwards applied to the daily precipitation at every grid point using the TFs that are common to all elements within the same group (or sub-group) and month. Thus…

**7.      Figure 3 is unclear. What do the boxplots represent and what is the true y-axis scale? Do the boxplots cover the spatial variability of monthly mean precipitation for the entire domain (a) or the elevation classes (b,c)? The text mentions that daily precipitation variability is shown, but how does this aggregate to monthly precipitation (y-axis label) then?**

*If boxplots really show the distribution of daily precipitation values does it really make sense to use the IQR? Depending on the wet day frequency more than 25% of the days might be dry, for instance.*

**RESPONSE**:

We appreciate that you bring to our attention that the y-axis, the caption and the text are confusing. To clarify this, we would like to mention that the boxplots illustrate the spatial distribution of monthly mean values of precipitation intensity across a specific area within 30 years. Thus, we have modified them as follows:

- The y-axis

Monthly precipitation [mm day$^{-1}$]

Precipitation intensity [mm day$^{-1}$]

- The text in the caption

Boxplots are illustrating the annual cycle and monthly distribution of daily precipitation: (a) entire Switzerland, (b) all grid points in the height class of 400 – 800 m, and (c) of 2.800 – 3.200 m. Black box-plots represent the observations (RhiresD data), blue and red ones the raw and corrected simulation, respectively. Top and bottom ends of the dashed lines represent the maximum and minimum values, respectively. Dots represent the mean.

Boxplots illustrate the spatial distribution of monthly mean values of precipitation intensity across a specific area within 30 years: (a) the area covers all grid points over entire Switzerland, (b) the grid points in the height class of 400 – 800 m, and (c) the grid points in the height class of 2.800 – 3.200 m. Black box-plots represent the observations (RhiresD data), blue and red ones the raw and corrected simulation, respectively. Top and bottom ends of the dashed lines represent the maximum and minimum values, respectively. Dots represent the spatial climatological mean value.

- Text, page 6 line 19 – 20 modified and moved to the beginning of the paragraph.

…, the annual cycle and the monthly distributions of daily precipitation are estimated for different height-classes …

…The annual cycle and the distributions of monthly mean precipitation intensity are for different height-classes to…

- Text, page 6 line 32 – 33

… For these example months, we present the patterns of biases in precipitation, changes in the distribution of daily precipitation, illustrated by the interquartile range as well as biases in wet-day frequency …

… For these example months, we present the spatial patterns of the biases in the monthly mean precipitation intensity, in the variability illustrated by the interquartile range, and in the wet-day frequency …

**8.      Also the general validation setup remains unclear to some extent, the validation technique and the respective reference datasets used needs to be better described. It is sometimes unclear whether the Swiss 2 km serves as reference or the Alpine 5 km grid.**

RESPONSE:

We agree that the validation technique and the data sets used are not fully described. To clarify it, we have modified it as follows:

- Page 5 lines 33 – 35 and page 6 lines 1 – 6

…To come up with a final method for the Alpine region we first test the influence of the different orographic characteristics (step 1). To be consistent with former studies (e.g., Sun et al., 2011; Themessl et al., 2012; Wilcke et al., 2013; Rajczak et al., 2016), the evaluation of the new method first uses the same region where the TFs are estimated. To be more rigorous, we additionally apply a cross-validation: Thereby, Switzerland is defined as the area to be corrected; then, we calculate two different TFs; namely, from the same Swiss region called Internal TFs (Int-TF), and from the corresponding Alpine region of Germany, France, and Austria altogether called External TFs (Ext-TF) (Fig. 1c). Note that Ext-TFs are carried out at 5 km horizontal resolution. To demonstrate the improvement of using the new method, we further compare it to a commonly used method that is carried out without orographic features and uses TFs deduced for the entire region of Switzerland (2 km) (similar to Berg et al., 2012; Maraun, 2013; Fang et al., 2015) …

… To come up with a final method for the Alpine region, we first evaluate the influence of the different orographic characteristics (step 1). To be consistent with former studies (e.g., Sun et al., 2011; Themessl et al., 2012; Wilcke et al., 2013; Rajczak et al., 2016), the evaluation uses the same region where the TFs are estimated. Explicitly, this means that the Swiss region in the WRF output (2 km) is defined as the area to be corrected and the RhiresD data set (at 2 km resolution) is used to obtain the TFs and to evaluate the different correction methods. These TFs are called Internal TFs (Int-TF) during the cross-validation process later on. Once the final method is determined, we additionally apply a cross-validation to test the method more rigorously: Thereby, Switzerland is defined as the area to be corrected (WRF output at 2 km resolution); in addition to the Int-TF (see above), which uses the same region to define TFs and to apply the correction, we also calculate a second set of TFs. The second one is obtained from the corresponding Alpine region of Germany, France, and Austria altogether called External TFs (Ext-TF) using the APGD data set (at 5 km resolution; Fig. 1c). Note that Ext-TFs are carried out at 5 km horizontal resolution and applied to Switzerland at 2 km resolution. To demonstrate the improvement of using the new method, we further compare it to a simple method that is carried out without orographic features and uses TFs deduced for the entire region of Switzerland (at 2 km resolution, 12 TFs in total) …

**9.      Any kind of bias correction will only be as good and as appropriate as the observational reference. The validity of an analysis of elevation dependencies and slope**

*dependencies at regional scales in the gridded observational precipitation datasets needs to be discussed. Does the reference grid really represent such dependencies?*

**RESPONSE**:

We appreciate this comment. We agree that we missed to show the validity of the elevation and slope dependencies in the gridded observational data sets. Note that the observational data sets have a height dependence on its quality. As mentioned by (Isotta, 2014), the gridded observational data sets do not only present errors due to the interpolation methods, but they also show errors that may differ in quantity from one to the other station (Sevruk, 1985; Richter, 1995) and are related to the "gauge undercatch", whose magnitudes range from 5% over the flatland regions to 30% above 1500 m a.s.l..To clarify this, a discussion is presented in results section of the revised manuscript.

Note that the observational data sets are considered generally reliable and represent orographic features well, although at high altitudes less data sets are available (Fig. R2; Isotta et al. 2014). Note that in this study we do not explicitly consider any uncertainty, and instead assume that these observations represent the true precipitation without errors. Still, we have discussed the uncertainty issue in particular for the results in high altitudes.

[Figure]

Figure R2. Swiss stations are integrated in RhiresD.

**10.     *The application of the Ext-TFs mixes spatial scales (classes based on 5 km orography vs. classes based on 2 km orography). This is potentially dangerous and the effects of this mismatch should be shown. Why is the validation, in this case, not carried out on the 5 km scale as well?***

**RESPONSE**:

We thank you for highlighting this point. To clarify it, we would like to mention that the method uses different observational data sets. We used the 5 km classes applied to 2 km target as we directly compare the results with the ones obtained from the application of Int-TFs and to avoid any additional uncertainty produced by interpolating between the two grids. Another reason is that the application at 5 km show minimal differences on the results, as is shown in the next

Figure R3. Therefore, we have mentioned this experiment in the results part of the revised manuscript but its figures are not shown because of the minimal differences.

[Figure]

R3. Biases in the climatological mean value of precipitation intensity over Switzerland. (a) represents the original biases in January, (b) the biases after being corrected at 5 km using Ext-TFs in January, (c) the biases after being corrected at 2 km using Ext-TFs in January, (d), (e), and (f) as (a), (b), and (c) but in July, respectively.

**Minor points:**

**a)** *page 1 line 19: "is" instead of "has been"*

We thank the reviewer for the suggestion. We have changed it in the manuscript.

**b)** *page 2 line 20: What is meant by "weaker intensity" here? Unclear*

It means that the simulated precipitation intensity is weaker than the observational one. As an example, instead of 20 mm day$^{-1}$ the simulated precipitation intensity is 5 mm day$^{-1}$. To make this point clear, we have modified it as follows.

… with a weaker intensity …

… with a lower intensity …

**c)** *page 2 lines 16-19: Line of argumentation unclear. RCMs were already referred to just above (line 12ff)*

We agree and to make the argumentation clearer, we have re-structured the paragraphs as also suggested in Major point #3 and the change is presented in the revised manuscript.

**d)** *page 4 lines 1-2: No true in general. Ban et al. for instance show that mean precipitation can also be much worse in convection resolving experiments. Certain aspects (such as the diurnal cycle) are improved, but not all.*

We agree that the statement in these lines is not in general true. To correct it, we have modified it as follows:

…Convection permitting model resolutions are preferred as recent studies show a better performance in simulating precipitation (e.g., Ban et al., 2014; Prein et al., 2015) …

…Convection permitting model resolutions are in general preferred as many recent studies show a better performance in simulating precipitation (e.g., Ban et al., 2014; Prein et al., 2015; Kendon et al., 2017; Berthou et al., 2018; Finney et al., 2019). However, we shall keep in mind that some biases in temperature and cloud formation may be produced by this set up, which may lead to additional biases in precipitation as shown in Ban et al. (2014) …

**e)** *page 4 lines 4-7: I don't really understand the reason behind this splitting in ten single 3-year simulations. 2 months spin up is certainly not enough for soil parameters and snow. Some more information on the setup and on the rationale behind it needs to be provided.*

Splitting up the simulations can be explained by the time-consuming setup to run a simulation over the Alps at 2 km resolution over 30 years. Namely, 3 model years are equivalent to 1 month in real time, which means that a 30-years simulation in a single piece would have taken at least 10 months in real time without any interruption.

Regarding the spin-up, we would like to mention that WRF has only an atmospheric component that is fed by initial and boundary conditions obtained from the GCM. Moreover, we consider the ice cover and soil in a quasi-stable state, as they are initially provided by the GCM and because of its long simulation these variables are in equilibrium there and because the interactions with the atmosphere are fully parametrised in WRF. Thus, the spin-up time was considered only for the atmosphere, which requires a much shorter spin-up period that certainly does not exceed two months.

**f)** *page 4 lines 19-20: I guess this is hardly true. In areas where no observations are available gridded products can be subject to very high uncertainties as inter- and extrapolation are required here.*

We agree that gridded products can be subject to important uncertainties in areas where there is no observation. To avoid misunderstandings, we have modified on page 4 the lines 18 – 20 as follows:

…The observational gridded data sets provide valuable insights, in particular in areas where observations are not possible due to extreme weather conditions or insufficient accessibility, such as mountain peaks. However, they also contain some discrepancies and uncertainties, e.g., high precipitation intensities are systematically underestimated and low intensities overestimated. …

…The observational gridded data sets provide valuable insights. However, they also contain some discrepancies and uncertainties due to inter- and extrapolation methods, e.g., high precipitation intensities are systematically underestimated and low intensities overestimated, especially in areas where observations are not available …

**g)** *page 5 lines 4-9: It remains unclear how these classes are computed. Based on the relation of a grid cell to its 8 direct neighbour grid cells? Please clarify.*

We thank you for bringing to our attention that this parameter remains unclear. To make it clear, we would like to mention that the slope-orientation is obtained by a simple trigonometric function using the two variables that are directly calculated by WRF. Namely, we sum two vectors: the slope north-south vector and the slope west-east vector, which both come directly from WRF. Thus, we have added additional information in the manuscripts follows:

- Page 5 line 8

    …< 315). Note that this characteristic is obtained by summing the two slope vectors that are directly provided by WRF. Combining …

**h)** *page 5 lines 15-17: Which threshold is then used in the present work?*

The threshold varies from group to group (or sub-group to sub-group) and from month to month. See major point 5.

**i)** *page 7 lines 30-32: This explanation seems to be not very likely given the turnaround time of atmospheric water vapor (a couple of days only). Water vapor should also frequently be resupplied by the boundary forcing of the RCM. Can you back this up by some reference?*

We appreciate that you bring this point to the discussion and we agree that the explanation needs to be improved. To achieve that, we would first like to mention that the drizzle effect is mainly caused by the horizontal resolution and the physics in the model (e.g. Gutowski et al. 2003; Chen and Dai 2019), and it can be independent of resupplying by the boundary conditions. Moreover, we have modified the explanation as follows:

… wet-day frequency may also explain the underestimation of the extreme precipitation (Fig. 3) as moisture necessary for extreme precipitation events is removed via the drizzle effect …

…wet-day frequency may slightly contribute to the underestimation of the extreme precipitation (Fig. 3) as precipitable water necessary for extreme precipitation events is removed via the drizzle effect. Namely, the precipitable water available for a daily extreme precipitation event may be distributed over several days due to problems in the parameterisations of the cloud microphysical and precipitation processes as found in Knist et al. (2018). …

Chen, Di, and Aiguo Dai. 2019. 'Precipitation characteristics in the Community Atmosphere Model and Their Dependence on Model Physics and Resolution'. Journal of Advances in Modeling Earth Systems 11 (7): 2352–74. https://doi.org/10.1029/2018MS001536.

Knist, Sebastian, Klaus Goergen, and Clemens Simmer. 2018. 'Evaluation and Projected Changes of Precipitation Statistics in Convection-Permitting WRF Climate Simulations over Central Europe'. Climate Dynamics, February. https://doi.org/10.1007/s00382-018-4147-x.

Gutowski, William J., Steven G. Decker, Rodney A. Donavon, Zaitao Pan, Raymond W. Arritt, and Eugene S. Takle. 2003. 'Temporal–spatial scales of observed and simulated precipitation in Central U.S. climate'. Journal of Climate 16 (22): 3841–47. https://doi.org/10.1175/1520-0442(2003)016<3841:TSOOAS>2.0.CO;2.

**j)**    *Figure 1: Why are Italy and Slovenia excluded from the Ext-TF analysis? They are part of the APGD dataset.*

Italy and Slovenia are excluded from the Ext-TF because of their poor station density covering the period 1979 – 2008 compared to the ones we used, especially over a complex topography and at high altitudes. This poor density could lead to more uncertainties in the dataset when representing the precipitation over complex topography, which could diminish the ability of the correction method. Therefore, we have included an explanation about this in the models and data section of the revised manuscript.

To clarify this, we show here two figures published in the website of Meteoswiss and in Isotta et al. (2014), respectively (Fig R4 and R5). Figure R4 and R5 show the station density used for creating the APGD data set. Moreover, Figure R4 presents the altitude of each station and Fig. R5 the time-covering fraction of the period 1971–2008 (Isotta et al. 2014).

[Figure]

Figure R4. Each point corresponds to a rain-gauge station for which data was available in the the spatial analysis. The color is the height (m) of the station. Source: https://www.meteoswiss.admin.ch/home/search.subpage.html/en/data/products/2015/alpine-precipitation.html)

[Figure]

Figure R5. Distribution of stations from which records of daily precipitation are integrated in APGD dataset. Shading represents the fraction of the full period (1971–2008) covered by the respective record. (Isotta et al. 2014)

**k)      Figures 4 and 5: Sorry, but it is unclear to me which bias is shown in these two figures. Bias of the IQR of daily precipitation amount sin Figure 5? Which intensity in Figure 4? Mean wet day intensity? Needs to be better explained.**

To clarify that, we have modified the captions of the three Figures as follow:

-   Figure 4

Biases of precipitation in terms of intensity over Switzerland. (a) represents the original biases in January, (b) the biases after being corrected using Int-TFs in January, (c) the biases after being corrected using Ext-TFs in January, (d), (e), and (f) as (a), (b), and (c) but in July, respectively.

Biases in the climatological mean value of precipitation intensity over Switzerland. (a) represents the original biases in January, (b) the biases after being corrected using Int-TFs in January, (c) the biases after being corrected using Ext-TFs in January, (d), (e), and (f) as (a), (b), and (c) but in July, respectively.

-   Figure 5

Biases of precipitation in terms of interquartile range over Switzerland. (a) represents the original biases in January, (b) the biases after being corrected using Int-TFs in January, (c) the biases after being corrected using Ext-TFs in January, (d), (e), and (f) as (a), (b), and (c) but in July, respectively.

Biases in the interquartile range of monthly mean precipitation intensity over Switzerland. (a) represents the original biases in January, (b) the biases after being corrected using Int-TFs in January, (c) the biases after being corrected using Ext-TFs in January, (d), (e), and (f) as (a), (b), and (c) but in July, respectively.

-   Figure 6

Biases of precipitation in terms of wet-day frequency over Switzerland. (a) represents the original biases in January, (b) the biases after being corrected using Int-TFs in January, (c)

Biases in the wet-day frequency within the 30-year period over Switzerland. (a) represents the original biases in January, (b) the biases after being corrected using Int-TFs in January, (c) the biases after being corrected using Ext-TFs in January, (d), (e), and (f) as (a), (b), and (c) but in July, respectively.

**Response to Referee #2**

**Specific comments:**

**1.      *The setup for the EMQ is completely unclear. Standard EQM is local, i.e. it would apply a different correction for each location for which observations are available, in this case for each gridcell of the observational datasets. There is no explanation of how the corrections for the subclasses (elevation and slope) are obtained. Are the local corrections averaged, or is the precipitation averaged prior to fitting the EQM?***
*This is obviously a key aspect of the method and it is surprising that it is not explained.*
*The statement that standard bias correction methods do not include the effect of topography is wrong, as the observations, which are the basis for the fitting, do include these effects. What is presumably meant is that standard bias correction does not include these effects explicitly, which means it cannot be applied when the topography changes*

**RESPONSE**:

We agree that the setup of the bias correction remains unclear. Still, we would like to point out that one strength of our method is that it is not local (the standard EQM described by the reviewer is a bias correction plus statistical downscaling). The simple reason is that a localized correction would fail in different states like the Last Glacial Maximum as valleys are filled with ice. To make the suggested method clearer, we have modified the manuscript as follows:

-   Page 5 lines 31 – 32

…To combine all steps, the EQM is applied to each (sub-) group and each month of the year, separately. This results in a set of TFs for each (sub-) group and each month of the year. Thus…

…To combine all steps, the local intensity scaling method and the EQM are applied to each (sub-) group defined in the first step and to each month of the year, separately, by pooling all grid points that belong to each group and handling them as a single distribution of daily precipitation. This results in a set of TFs for each (sub-) group and each month of the year. For instance, when the correction is carried out using height-classes of 400 m, a TF is defined for each group, resulting in nine TFs for each month and in total 108 TFs throughout the year. Moreover, the correction is afterwards applied to the daily precipitation at every grid point using the TFs that are common to all elements within the same group (or sub-group) and month. Thus…

We also agree that the observational data sets implicitly include effects of topography. Changes regarding this point are presented in the following lines of the manuscript:

-   Page 2 line 33

…correction methods do not consider orographic features that…

…correction methods only implicitly consider orographic features that…

- Page 3 line 6

…time includes orographic characteristics…

… explicitly combined orographic characteristics…

**2.    *As already pointed out by the first reviewer, determining joint bias corrections for the subclasses defined by topography and slope only makes sense if the local bias corrections within a class are more similar than those between the classes. This needs to be shown***

**RESPONSE**:

We appreciate this comment and we agree that we missed to show clearly enough the argumentation for using different classes. As reviewer 1 asked a similar question we present there the same answer: We thank the reviewer for bringing up this concern. We agree that the main purpose of the correction method might still be a bit unclear and we would like to clarify this in more details in the following. With the present study, we would like to obtain a flexible correction that can be applied to several different climate states at the same time. To obtain this, the correction method should not be constrained to the actual climate too much, this is, because circulation changes and atmospheric characteristics may be variable between different climates. We agree that a cluster analysis of precipitation and its errors should be applied, so that errors can be grouped accordingly and to keep the error within classes as small as possible, to obtain an optimal correction result. This has for example been performed by Gomez et al. (2018) for Switzerland. The drawback of such a correction for our purpose is that such a cluster analysis is always based on the characteristics and circulation of the current climate and this is what we would like to avoid as much as possible. To be as much independent from current climates as possible and to still provide a correction that still touches upon important characteristics in the Alpine climate, we came up with "static" characteristics, i.e. topography height and orientation. Both, topography and orientation will remain similar during different climate states, even if we are aware of the fact that in any correction the effect of topography is implicitly included. Nevertheless, we would like to show here that biases have some orographic dependence. To clarify this, we have attached a figure that presents the monthly mean biases for each height-class before and after the correction (Fig. R1). Figure R1 illustrates an overestimation at high elevations and an underestimation at the lower ones during the colder months. Moreover, different levels of underestimation are observed across the height-classes during the warmer months. Thus, the splitting into different height-classes is appropriate to be used in the bias correction. Moreover, we would like to mention that we explicitly present the model biases within two classes in the Fig. 3 (of the manuscript), and implicitly for all the height classes in Fig. 4 and 5. Note that the biases within the classes are much smaller than between the classes. Therefore, we have included a more balanced discussion about our approach in results section of the revised manuscript

[Figure]

Figure R6. Mean bias over Switzerland for different height-classes.

**3.**      ***The justification for the intended application is superficial and ignores key problems. In turn this means that the justification for the new approach itself is weak. As pointed out already by the first reviewer, many things in addition to the topography are different in a glacial climate, for instance the large-scale circulation or the moisture content. It is thus highly questionable whether applying a bias correction that is based on present climate, even if it explicitly accounts for topography, would yield meaningful results.***
***This problem is closely related to the distinction of different types of errors and to the issue of propagation of GCM errors through dynamical downscaling. There are a few statements in the paper that mention that discrepancies of RCM simulations and observations might be caused by the driving GCM. However, there is no systematic discussion of what kind of errors bias correction could correct in a meaningful way. A discussion of these issues can be found for instance in***
***Maraun, et al., 2017: Towards process-informed bias correction of climate change simulations. Nature Climate Change, 7(11), 764-773***
***Maraun and Widmann, 2018: Statistical downscaling and bias correction in climate research. Cambridge University Press, ISBN 1107066050***
***Eden, J.M., Widmann, M., Grawe, D, and Rast. S., 2012: Reassessing the skill of GCM-simulated precipitation. J. Climate, 25(11), 3970-3984.***

**RESPONSE**:

We appreciate that the reviewer brings up the point that it might be misleading to what extent the presented bias-correction can be applied to other climate states. As already responded to reviewer 1, we would like to mention that the danger of correcting biases in a simulated climate with a method that has been trained with a climate that does not correspond to the simulated one is well-known in the statistical downscaling and correction methods. Statistical downscaling and correction methods suffer basically from the assumption of stationary biases, which implies that their algorithms trained with today's climate are considered to be also valid for different climate states. Thus, our work aims at presenting a new bias-correction that attempts to decrease this danger by using orographic features, which are less likely characteristics of the current climate only. Moreover, precipitation biases are not only produced by initial and boundary conditions provided by the global climate models, but also by parametrisations, physical and numerical formulations that are described in both global and

regional climate models. The main goal of the presented work is to correct wet or dry biases that stem either from global or regional models or both. These biases can be produced by parametrisations and numerical formulations, but those that are mainly associated with orographic effects, namely, vertical motion leading to precipitation. To clarify this, we extended the discussion on the general shortcomings of bias correction methods in the conclusion section of the revised manuscript. Note that the presented correction is only applicable in regions where the topography is rather complex and where topography has certainly an influence on the local atmospheric circulation.

**4.** ***The fact that EQM leads to correct distributions for the fitting data is trivially true by construction. The informative part of the validation of statistical models is related to the aspects that are not trivially in agreement with observations. For each aspect of the validation it should be discussed to what extent a good skill can be expected by construction. For instance, given the unclear setup for fitting and application of the bias correction, it is not clear what causes the differences between observed and corrected distributions in Fig.3, or the differences in Fig. 4 and Fig. 5.***
***Some problems related to the validation of bias correction methods are discussed in Maraun, D. and M. Widmann, 2018, 'Cross-validation of bias-corrected climate simulations is misleading', HESS, 22(9), 4867-4873.***

**RESPONSE**:

We thank the reviewer for this comment and agree that the validation discussion can be improved. As noted by Bennett et al. (2014), the importance of cross-validation methods is that they can test the ability of bias-correction techniques on a different climate state. However, this might not be reasonable as the biases of the other climate state may not remain unchanged and the method's accomplishment relies on the biases caught during the period the method is trained on. We also recognise that recent studies by Maraun et al. (2017) and Maraun and Widmann (2018) have argued against carrying out a cross-validation for evaluating bias corrections. The authors remarked that the observational and simulated data sets do not have a synchronised internal climate variability. Thus, this asynchronism in the internal climate variability may be one of the sources of the biases in free-running models.

Furthermore, as mentioned by Maraun and Widmann (2018), our cross-validation method does not compare the correction to the observations on the validation period (future or past climate state), which can produce false positive or true negative results due to internal variability in the model or observations, but the method assesses whether the statistical evolution of the model is kept.

Moreover, one of the reasons that may explain the remaining difference between the observational and the corrected data sets, as mentioned in the manuscript, can be traced back to the fact that some height classes sample over regions with slightly different biases. Hence, biases of one area could be diminished by the biases that are shared by the other areas. For instance, the strong negative biases observed in the Rhone Valley and Ticino are not fully corrected because the slight underestimation across the Swiss Plateau dominates the bias in this height-class.

Nevertheless, we agree that the evaluation and the argumentation for the remaining biases is not discussed clearly enough in the manuscript and that this should be better explained. Thus,

we have extended the discussion more explicitly in the results and conclusion section of next version of the manuscript.

Bennett, James C., Michael R. Grose, Stuart P. Corney, Christopher J. White, Gregory K. Holz, Jack J. Katzfey, David A. Post, and Nathaniel L. Bindoff. 2014. 'Performance of an empirical bias-correction of a high-resolution climate dataset'. International Journal of Climatology 34 (7): 2189–2204. https://doi.org/10.1002/joc.3830.

**5.        *It is not clear why the wet-day frequency is adjusted prior to the fitting of the EQM. If EQM is applied to the whole distribution including dry days, this adjustment is included in the EQM fitting. The justification might be linked to the unexplained details in the fitting setup.***

**RESPONSE**:

We thank the reviewer for highlighting this point and recognize that this adjustment may not be clear enough. We would like to mention that the adjustment does not mainly focus on the wet-day frequency, but the very low intensity values. As clarified already in the answer for reviewer 1, we agree that the argumentation for this adjustment can be better explained. To make this clear, we would like to mention that, in our study, we use an empirical quantile mapping technique (EQM) that differs from the parametric quantile mapping technique (QM). The reason of using an EQM is because this technique uses an empirical cumulative distribution function and does not fit any parametric distribution to the sample, i.e, (sub-) groups, as it is done in the QM. Therefore, we do not assume any known distribution either in our data sets or in the possible application to other climate states. However, the results of the EQM can become unrealistic if the very low intensity values are not adjusted previously. The reason for this is that these values can produce inappropriate TFs due to an important shift in the distribution, i.e., the quantiles (Teutschbein and Seibert, 2012; Lafon et al. 2013).

To adjust these very low values, an additional parameter is included in the definition of days without precipitation that has been mentioned before in the respond of the second major point of reviewer 1. The days without precipitation are not considered for calculating the TFs when they fall below a certain threshold. Many studies use a static threshold for entire data set that is between 0.01 and 1.00 mm day$^{-1}$, whereas in our study, we calculate a static threshold for each group (or subgroup) and months of the year. This allows to be the consistent with the different biases-treatment across the groups (or subgroups) and months of the year. The threshold is calculated using the local intensity scaling method and can vary vary in our study from 0.001 to 1.00 mm day$^{-1}$. To clarify this, we have made some changes that are presented in the revised manuscript and also in response to the fifth major comment of reviewer 1.

Changes in the manuscript are presented as follows:

-    Page 5 lines 13 – 14

…2010). To correct precipitation with very low-intensity the first part of the local intensity scaling method is used (Schmidli et al., 2006). It consists …

…2010), which can distort the precipitation distribution substantially, i.e., shifting the quantiles, producing inappropriate corrections in the third step when EQM is applied

(Teutschbein and Seibert, 2012). To correct precipitation with very low intensity, an additional parameter is included in the definition of dry days related with the uncorrected precipitation. Dry days are not considered for calculating the TFs when they fall below a certain threshold. Many studies use a static threshold for the entire data set which is between 0.01 and 1.00 mm day[1] (Piani et al., 2010a; Lafon et al., 2013; Maraun, 2013). We calculate a static threshold for each group (or subgroup) and months of the year. This allows to be the consistent with the different biases-treatment across the groups (or subgroups) and months of the year. Then, we carry out the local intensity scaling method (Schmidli et al., 2006) that is also used by Teutschbein and Seibert (2012) before using the quantile mapping technique. This method consists …

- Page 5 lines 16 – 17

…The threshold can vary from group to group, but it is often close to or smaller than 1 mm day[1] Schmidli et al., 2006).

…In our work, the threshold can vary from group to group and from month to month between 0.001 and 1 mm day-1 as in Schmidli et al. (2006) …

**6.      Although it is mentioned that the errors in the observations should be taken into account when interpreting the results, there is no substantial effort to actually do this. For instance, it would be instructive to do a rough correction for the substantial undercatch of precipitation falling as snow, which strongly affects the high elevations, and assess to what extent the validation results are sensitive to this error.**

**RESPONSE**:

We appreciate this comment. We agree that we missed to show a wider discussion about the error in the observational data sets when interpreting the results of the correction method. As reviewer 1 asked a similar question we present there the same answer: As mentioned by (Isotta, 2014), the gridded observational data sets do not only present errors due to the interpolation methods, but they also show errors that may differ in quantity from one to the other station (Sevruk, 1985; Richter, 1995) and are related to the "gauge undercatch", whose magnitudes range from 5% over the flatland regions to 30% above 1500 m a.s.l.. Therefore, we have included a better discussion of these errors when analysing the correction, which is presented in the results discussion part of the revised manuscript.

Sevruk B. 1985. Systematischer Niederschlagmessfehler in der Schweiz. Der Niederschlag in der Schweiz, Beitr̈age zur. Geologischen Karte der Schweiz-Hydrologie 31: 65–75.

Richter D. 1995. Ergebnisse methodischer Untersuchungen zur Korrektur des systematischen Messfehlers des Hellmann-Niederschlagsmessers. Bericht Deutschen Wetterdienstes 194, 93 pp. (To be obtained from German Weather Service, Offenbach a.M., Germany.)

**7.      As the realization of internal variability is different the observations and in a free-running GCM (as opposed to a reanalysis) some differences between observations and simulations will be due to internal variability. This effect should be roughly quantified, for**

*instance by showing fitting and validating the method for 10 or 15 year sub-periods (which would lead to 9 or 4 possible combinations of fitting and validation subperiods).*

**RESPONSE**:

We thank the reviewer for bringing to our attention the approach to quantify the biases that may be caused by differences between the internal variability of the observational data set and the simulated one. Furthermore, we would like to mention that correction methods are sensitive to the period the methods are trained on, and their accuracies would increase as more information from the observational data sets is taken into account (Lafon et al., 2013). Therefore, since the accuracy of our correction method needs to be kept as high as possible, we have carried out the suggestion made by the reviewer by splitting the data sets into two sub-periods, which is explained and analysed in the following paragraphs.

To quantify any difference that may be caused by using data sets with different internal variabilities, we have calculated two additional sets of Int-TFs using the first and last 15 years, separately. Note that we avoid shorter periods (like the suggested 10 yrs) as less data is available to estimate the TFs. Each set of Int-TFs is then applied to the 30-year simulated precipitation over Switzerland, to be comparable with the 30-yr period used so far in the manuscript. Thus, we obtain two newly corrected precipitation data sets (15yr-A and 15yr-B, respectively) that are compared to the data set that was obtained by the correction trained with the 30-year period (30yr). To assess the difference related to colder and warmer months, we select, as in the manuscript, two months that mainly represent each period; namely, January and July.

Focusing on the biases in the climatological mean value of precipitation intensity, and comparing the original biases with the three approaches, we observe that the methods carried out with 15yr-A and 15yr-B illustrate a correction similar to the method with 30yr. Namely, they reduce the overestimation over high mountain regions during colder months and the general underestimation during warmer months. In addition, the regions with remaining biases agree with the remaining biases of the correction with 30yr. Still, some differences between the 15yr-A and 15yr-B and the method using 30yr are evident: During January, the method using 15yr-A shows a better performance over the high altitudes but not over the flatlands and in the Ticino, and inversely, the method using 15yr-B outperforms the latter areas but not over the mountains (Fig. R7). During July, the method using 15yr-A outperforms over the flatlands and the Ticino but not the high altitudes, and inversely, the method using 15yr-B shows a better performance over the latter area but not over the flatlands and in the Ticino (Fig. R8). This demonstrates that the method calibrated with the two sub-periods can slightly influence the correction method but its effects can be considered minimal when the work by Lafon et al. (2013) is taken into account. As described before, Lafon et al. (2013) found that the accuracy of the correction methods is sensitive to the period the methods are trained on, which could explain some of the remaining biases when using 15yr-A and 15yr-B Therefore, we have mentioned this experiment in the results part of the revised manuscript but its figures are not shown due to the minimal effects.

[Figure]

Figure R7. Biases in the climatological mean value of precipitation intensity in January over Switzerland. (a) represents the original biases, (b) the biases after being corrected using Int-TFs obtained from the 30-year period, (c) as in (b) but from the first 15-year period, (d) as (c) but the second 15-year period.

[Figure]

Figure R8. Biases in the climatological mean value of precipitation intensity in July over Switzerland. (a) represents the original biases, (b) the biases after being corrected using Int-TFs obtained from the 30-year period, (c) and (d) as in (b) but Int-TFs obtained from the first and second 15-year period, respectively.

Once again, we would like to thank the reviewer for the time invested to review our paper so carefully and we are looking forward to meeting the reviewers' expectations.

Best regards,

Patricio Velasquez

[revised manuscript text omitted]

---

## Referee Report (RR1)

**Re-evaluation of the manuscript**

**A new bias-correction method for precipitation over complex terrain suitable for different climate states: a case study using WRF (version 3.8.1)**

**by Patricio Velasquez et al.**

**for publication in Geoscientific Model Development**

*General comments:*

The body of the manuscript in its present form describes an extension and modification of a bias correction method (Empirical Quantile Mapping, EQM) for a regional climate model with application over areas with complex terrain and past climatic periods with changed orography. The manuscript is well written and the method is described, validated and comprehensively discussed under present-day climate conditions. In the present form however, an important part of the analysis – the application and consistency test with Last Glacial Maximum (LGM) conditions and simulations – is still missing. This part should be presented in greater detail by adding a dedicated full chapter on the LGM simulations and according application of the method to assure the robustness and added value of the method.

*Abstract:*

A good and concise summary, but for the general audience maybe one or two sentences on some hypotheses that could be better addressed concerning the LGM-PD climatic differences. i.e. changes in human occupation and migration routes, implications for interpretation of hydro-sensitive proxies etc. to put the work into a broader context (even the empirical evidence is very sparse). The most important point, the application and presentation to LGM conditions and consistency checks with empirical data is still missing.

*Introduction*:

The introduction is quite technical towards the modelling side. Maybe the authors can add some general remarks on basic climate differences between present day and LGM, i.e. mean temperature differences. A zoom of the glaciated area over Switzerland would also be interesting to see in a plot.

In addition, some hints to proxy studies are missing, e.g. what is the benchmark and/or any hypotheses that could be addressed specifically with the study and the method. Those questions do not necessarily need to be all addressed in this study, but it would be important for the general audience to see an applicated and added value of the technical work of the bias correction.

Concerning the stationarity assumption: What are the most important challenges violating the stationarity assumption ? (e.g. differences in lapse rates Present day-LGM, influence of albedo on temperature profiles, general lower availability of moisture and precipitable water during glacial times, changes in circulation, importance of circulation and other biases of the driving model etc. how are changes in vegetation cover treated as an important source of moisture-recycling).

Please just mention those issues to put the precipitation bias correction into perspective in concert with other, competing and maybe evening cancelling factors, complicating the eventual validation on bias corrected results with data based on empirical evidence and other modelling approaches.

An interesting question for future studies that might be mentioned is how the changed precipitation itself could alter maybe even large-scale climate in terms of soil moisture and/or snow-albedo effects that cannot be accounted for after the bias correction is applied.

*Model and Data:*

The general setup seems to be very innovative and is an original approach for this time slice by implementing a simulation with 4x4km over the highly complex terrain over the alpine region.

A very meaningful implementation in the setup is the very conservative downscaling factor of 1:3, addressing the different meso-and local scale features that cannot be reproduced using a more liberal conversion strategy by leaving out some nests.

*Validation of the method:*

For present day climate conditions, the method is very rigorously tested and testing/validation is reported in a very detailed and concise way. Unfortunately, the LGM (with altered topography forming the test bed and a central part of the study) is only hardly addressed in a very short last paragraph. The added value, also leaving out the LGM, seems to be the height-class dependent quantile mapping for present day climate. For the reader, however, the direct comparison to a classical EQM is missing in Figures 5–7. This analysis would be helpful in order to assess the advantage of the new bias correction method.

*Conclusions:*

The conclusions mostly pertain to situation of present-day with unchanged topography. The LGM is only hardly presented, although focus is set on changes in topography. Therefore it would be really important to show more results of the LGM. In the present form, the conclusions present more or less a repetition and summary of the validation of the methods section.

I suggest adding a dedicated results section on the added value including more material of the LGM simulations and for comparison at least some studies based on empirical evidence. In the present form I think the manuscript presents too little original and robust information/analysis that the method is outperforming classical EQM methods to qualify it as a comprehensive scientific paper.

*Additional references and sources for comparisons with LGM*

Allen, R., Siegert, M. J., and Payne, A. J.: Reconstructing glacier-based climates of LGM Europe and Russia – Part 2: A dataset of LGM precipitation/temperature relations derived from degree-day modelling of palaeo glaciers, Clim. Past, 4, 249–263, https://doi.org/10.5194/cp-4-249-2008, 2008.

Becker, P., Seguinot, J., Jouvet, G., and Funk, M.: Last Glacial Maximum precipitation pattern in the Alps inferred from glacier modelling, Geogr. Helv., 71, 173–187, https://doi.org/10.5194/gh-71-173-2016, 2016.

Simon Brewer, Thomas Giesecke, Basil A. S. Davis, Walter Finsinger, Steffen Wolters, Heather Binney, Jacques-Louis de Beaulieu, Ralph Fyfe, Graciela Gil-Romera, Norbert Kühl, Petr Kuneš, Michelle Leydet & Richard H. Bradshaw (2017) Late-glacial and Holocene European pollen data, Journal of Maps, 13:2, 921-928, DOI: 10.1080/17445647.2016.1197613.

Florineth, D., & Schlüchter, C. (2000). Alpine Evidence for Atmospheric Circulation Patterns in Europe during the Last Glacial Maximum. *Quaternary Research, 54*(3), 295-308. doi:10.1006/qres.2000.2169

Luetscher, M., Boch, R., Sodemann, H. *et al.* North Atlantic storm track changes during the Last Glacial Maximum recorded by Alpine speleothems. *Nat Commun* **6,** 6344 (2015). https://doi.org/10.1038/ncomms7344.

---

## Referee Report (RR2)

**Final comments on the manuscript**

**A new bias-correction method for precipitation over complex terrain suitable for different climate states: a case study using WRF (version 3.8.1)**

**by Patricio Velasquez  et al.**

**for publication in Geoscientific Model Development**

*General:*

The authors addressed in a concise and comprehensive manner the points raised in the former evaluation and implemented suggestions to improve and extend their manuscript. Moreover, results of the last Glacial Maximum simulation are included in the new version. I also would like to emphasize the very detailed answers to the points raised in the former evaluation, especially to clarify some potential misunderstandings.

Below I list some minor comments and suggestions that might be helpful to prepare the final version of the manuscript for final publication.

*Minor comments and suggestions:*

*Abstract*:

p.1 l.4          please add […human occupation and *according migration routes*].

*1 Introduction*:

p.2 l.20          please change: [ … these tools provide *physically consistent* and spatially gridded three dimensional information on various meteorological variables. ]

p.3 l.15          please add that also temperature thresholds are important as absolute values (e.g. limiting factor for vegetation coverage, freezing of water, snowfall vs. liquid precipitation)

p.3 l.21          please reformulate: [ If the method is applied to ….]

p.4 l 4          please reformulate: […that can ameliorate the stationarity assumption…]

*2 Models and Data:*

p.5 l. 20          please add in the text which kind of convection scheme you are using for the convection-permitting simulations (with reference to Table 2)

*3 Bias correction:*

p.7 l.24          I suggest to put the formulas for Q_SIM* and Q_OBS into dedicated lines with numbering of formulas for a quicker reference for the reader

*4.1 Biases of WRF and their seasonality:*

p.9 ll.3 ff          I suggest to include the scatter diagrams as separate figures or subfigures into the figure section

p.10 l.3          I was wondering whether the vast glaciation of the LGM over the alpine region has an impact on near-surface condensation processes, i.e. freezing of near-surface moisture on the surface of the ice and whether those processes are addressed in the WRF model. This might be listed as another source of uncertainty that can only hardly be quantified, but might play a role when bias correcting results for LGM-type of climates.

*4.2 Influence of different orographic characteristics on the performance of the bias-correction method:*

p.11 l.14          A second issue explaining the somewhat better performance of the 100 m interval relates to the better fit of the transfer function in the respective height interval – in complex alpine regions a vertical difference of 400 m can be climatologically quite large compared to the one using only 100 m.

*4.3 Application of the bias-correction method and cross-validation under present-day conditions:*

p.11 l.25          please re-formulate: [ We consider… → A priori, this comparison is based on different prerequisites, as…]

p.12 l.4          please replace "works" by "is appropriate"

p.12 l.24 ff          In this context, 15 years in each sub-period might not cover the full range of the internal climate variability. I assume that longer validation/verification intervals ameliorate this effect [ as the authors indirectly state on p. 13 l.2. ] An alternative is for instance to apply a one-leave-out type of method to hold back most of the years for calibration. In this context in my opinion, a qualitative statement would be enough as additional background information for the reader.

*4.4 Application of bias-correction methods on the simulated LGM climate:*

p.16 l.6          please change "safer" to "is better suited"

*5 Summary and conclusions:*

One might also put the method into context of future climate change: the current example is on LGM with a potentially reduced complexity in terrain due to vast glaciers. In future scenarios this might be the opposite, especially over areas with presently still extensive glaciated areas like the Himalayas Mountains. In those areas increased melting of glaciers increases the complexity of the terrain and hence the application of according bias correction methods might become important.

*Figures:*

General:     If possible, please indicate in each Figure caption the main conclusion of the Figure(s) as immediate summary for the reader

Fig. 1:       In Fig. 1b the lat/lon information is missing. Also for Fig. 1c it would be desirable to diplay separately the map including lat/lon information and the legend

Fig. 3, caption:  I suggest keeping the formulation "Empirical Quantile Mapping (EQM)"

---

## Author Response (AR3)

**Response to Editor Comment**

Dear Dr. Fabien Maussion,

We greatly thank you for the time dealing with our manuscript, especially the careful and thorough reading. Your comments and suggestion have been carefully considered.

To make our study much more valuable, we decided to include an analysis about the applicability of the new method to other climate state with strongly changed topography, i.e., the LGM climate. As mentioned in a previous response, our method shows a smoother bias-correction for the LGM; whereas the major valleys of Switzerland are visible in the standard EQM. The latter is questionable as the valleys are filled with ice during the LGM. To be more precise in our purpose, the manuscript is further extended. Please see the marked-up manuscript.

Once again, we highly appreciate your comments and suggestions to the manuscript and we are looking forward to meeting your expectations.

Best regards,

Patricio Velasquez

[revised manuscript text omitted]

---

## Author Response (AR4)

**Final Response to Referees**

We thank the reviewer for the careful and thorough reading of our manuscript. The comments have been carefully considered and responded. Please find below our response to each comment.

*1.      General comment: In the present form however, an important part of the analysis – the application and consistency test with Last Glacial Maximum (LGM) conditions and simulations – is still missing. This part should be presented in greater detail by adding a dedicated full chapter on the LGM simulations and according application of the method to assure the robustness and added value of the method.*

**RESPONSE**:

We included a dedicated subsection on the LGM. This subsection contains a short description of the LGM precipitation pattern in the raw simulation. Additionally, we assess the application of the standard EQM and the new bias-correction methods to the simulated LGM climate. This provides more details about their performances and the added value of the new method. Note that we also followed the suggestions of the reviewer and highlighted results of the LGM simulation in the abstract, made a brief introduction of the LGM in section 1 and extended the conclusions on the LGM.

*2.      Abstract: A good and concise summary, but for the general audience maybe one or two sentences on some hypotheses that could be better addressed concerning the LGM-PD climatic differences. i.e. changes in human occupation and migration routes, implications for interpretation of hydro-sensitive proxies etc. to put the work into a broader context (even the empirical evidence is very sparse). The most important point, the application and presentation to LGM conditions and consistency checks with empirical data is still missing.*

**RESPONSE**:

Thank you for this comment. We implemented a better motivation in the abstract and also mention the results under LGM conditions.

*3.      Introduction: The introduction is quite technical towards the modelling side. Maybe the authors can add some general remarks on basic climate differences between present day and LGM, i.e. mean temperature differences. A zoom of the glaciated area over Switzerland would also be interesting to see in a plot.*

**RESPONSE**:

Clearly, the main purpose of the study is to present the new bias-correction method, so the technical scope of the introduction is given. However, as we decided to follow the suggestion

of the reviewer to give more weight to the LGM results we also think that a brief description of the differences between present-day and LGM conditions in the introduction of the revised manuscript is beneficial. We also added crosshatched areas in Fig. 2b to represent the Alpine glacier coverage (ice cap) used in the LGM simulation. Note that this ice cap corresponds to the 21-kya slice of the simulation performed by Seguinot et al. (2018). We referenced this ice cap distribution in the description of the simulations.

Seguinot, Julien, Susan Ivy-Ochs, Guillaume Jouvet, Matthias Huss, Martin Funk, and Frank Preusser. 2018. 'Modelling Last Glacial Cycle Ice Dynamics in the Alps'. *The Cryosphere* 12 (10): 3265–85. https://doi.org/10.5194/tc-12-3265-2018.

**4.      Introduction: In addition, some hints to proxy studies are missing, e.g. what is the benchmark and/or any hypotheses that could be addressed specifically with the study and the method. Those questions do not necessarily need to be all addressed in this study, but it would be important for the general audience to see an applicated and added value of the technical work of the bias correction.**

**RESPONSE**:

We thank the reviewer for this comment. As for the abstract, we added some potential application fields of the method. Certainly, the method is needed where absolute values of precipitation are used. An example is glacier modelling. Others are assessing human occupation and pathways, where water availability was an important parameter. We also summarize studies which do not only define the LGM climate, but also compare some global and European characteristics to present-day conditions (see new paragraph in the introduction).

**5.      Introduction: Concerning the stationarity assumption: What are the most important challenges violating the stationarity assumption? (e.g. differences in lapse rates Present day-LGM, influence of albedo on temperature profiles, general lower availability of moisture and precipitable water during glacial times, changes in circulation, importance of circulation and other biases of the driving model etc. how are changes in vegetation cover treated as an important source of moisture-recycling). Please just mention those issues to put the precipitation bias correction into perspective in concert with other, competing and maybe evening cancelling factors, complicating the eventual validation on bias corrected results with data based on empirical evidence and other modelling approaches.**

**RESPONSE**:

We thank the reviewer for the helpful comment. We certainly have these processes in mind which violate the stationarity assumption, but we think it is an excellent idea to mention some of them explicitly in the introduction, so the reader is aware of them.

***6.     Introduction: An interesting question for future studies that might be mentioned is how the changed precipitation itself could alter maybe even large-scale climate in terms of soil moisture and/or snow-albedo effects that cannot be accounted for after the bias correction is applied.***

**RESPONSE**:

We think that the reviewer suggests a very interesting question. We consider this in the conclusions of the paper as with our study we cannot solve this issue.

***7.     Method and Data: The general setup seems to be very innovative and is an original approach for this time slice by implementing a simulation with 4x4km over the highly complex terrain over the alpine region.***

**RESPONSE**:

We thank the reviewer for this comment, we would like to point out that we use an even finer resolution of 2x2 km over the Alpine region.

***8.     Validation of the method: For present day climate conditions, the method is very rigorously tested and testing/validation is reported in a very detailed and concise way. Unfortunately, the LGM (with altered topography forming the test bed and a central part of the study) is only hardly addressed in a very short last paragraph. The added value, also leaving out the LGM, seems to be the height-class dependent quantile mapping for present day climate. For the reader, however, the direct comparison to a classical EQM is missing in Figures 5–7. This analysis would be helpful in order to assess the advantage of the new bias correction method.***

**RESPONSE**:

To better highlight the results under LGM climate and to show the strength of the newly presented bias-correction, we have indeed added a new subsection. We think the reviewer is correct in suggesting to emphasize the LGM a bit more in the manuscript, as it is the one climate state, where our bias-correction should be able to add value compared to the standard EQM.

Concerning Fig. 5-7, we believe that there might be some misunderstandings. We do not intend to infer that the new height-dependant bias-correction is better than the standard EQM in the present-day climate. To recall, the standard EQM adjusts the simulated precipitation distribution to match the observed one at a grid-point level. This makes the standard EQM more than just a bias-correction method, the standard EQM also functions as statistical downscaling method. In cases when the training and application period are the same, the standard EQM would assuredly outperform all methods that do not correct at this grid-point level. This is because all mean biases are pointwise removed in the standard EQM. Consequently, Fig. 5-7 would be mostly white as the standard EQM outperforms the new

method. Thus, we argue that such a comparison would be unfair, given that the standard EQM is also a statistical downscaling. Hence, we do not think that it is meaningful to add the standard EQM in Fig. 5-7 applied to the present-day climate as it would be almost white, as explained above.

For LGM conditions, this is different. Here, a comparison between the new method and the standard EQM certainly makes sense. The reason is that the strength of the standard EQM (correction at grid-point level) under present-day climate might be a weakness under highly different climate states, since local-related biases might not exist. An example is already mentioned in the manuscript; namely, the Rhone valley is a narrow deep valley with highly localized present-day climate conditions and thus a very specific bias structure. Whereas the valley is mostly filled with ice during the LGM and does not exist in the present-day form. Thus, the statistical downscaling of the standard EQM would apply an invalid correction as the specific local present-day climate conditions do not exist during the LGM.

We more explicitly reformulated and added some sentences in different parts of the revised manuscript to clarify that the standard EQM outperforms our method under present-day conditions due to the double strategy bias-correction and statistical downscaling. Also, we hope that this comment is satisfactorily resolved with the added subsection on the LGM simulation.

***9.      Conclusion: In the present form, the conclusions present more or less a repetition and summary of the validation of the methods section.***

**RESPONSE**:

We thank the reviewer for this comment. We agree that the conclusion section also present a brief summary of the study. To better match the section's title with its content, we changed the title to "Summary and conclusions". In addition, we adjusted the conclusion on the LGM experiment and briefly extended the outlook.

***10.      Conclusions: I suggest adding a dedicated results section on the added value including more material of the LGM simulations and for comparison at least some studies based on empirical evidence.***

**RESPONSE**:

As suggested, we added a new paragraph dedicated to the LGM climate and the application of the bias-correction to it.

***11.      In the present form I think the manuscript presents too little original and robust information/analysis that the method is outperforming classical EQM methods to qualify it as a comprehensive scientific paper.***

**RESPONSE**:

We thank the reviewer for the helpful and highly appreciated comments, i.e., enlarging the motivation, given more weight to the LGM results, etc. We think that with these modifications and additions we are able to show the originality and robustness of the new method and the need for such a method to answer paleo-climatic research questions.

Once again, we would like to thank the referee for the time invested to review our manuscript so carefully and we are looking forward to meeting his/her expectations.

Best regards,

Patricio Velasquez

[revised manuscript text omitted]

---

## Author Response (AR5)

**Final Response to the Referee**

We thank the reviewer for the time invested to review our manuscript so carefully and we are looking forward to meeting his/her expectations.

The suggestions have been carefully considered and we have followed the corrections made by the reviewer. The only point we do not agree is to add the main conclusion in the captions of the figures, since the guidelines of the journal asks for concise and descriptive captions.

Please find the changes according to the suggestions in the revised version of the manuscript.

Best regards,

Patricio Velasquez

[revised manuscript text omitted]